# STEM enables mapping of single-cell and spatial transcriptomics data with transfer learning

Minsheng Hao [1], Erpai Luo[1], Yixin Chen [1], Yanhong Wu [1], Chen Li[1], Sijie Chen[1], Haoxiang Gao [1], Haiyang Bian[1], Jin Gu [1], Lei Wei [1✉] & Xuegong Zhang [1,2✉]

Profiling spatial variations of cellular composition and transcriptomic characteristics is important for understanding the physiology and pathology of tissues. Spatial transcriptomics (ST) data depict spatial gene expression but the currently dominating high-throughput technology is yet not at single-cell resolution. Single-cell RNA-sequencing (SC) data provide high-throughput transcriptomic information at the single-cell level but lack spatial information. Integrating these two types of data would be ideal for revealing transcriptomic landscapes at single-cell resolution. We develop the method STEM (SpaTially aware EMbedding) for this purpose. It uses deep transfer learning to encode both ST and SC data into a unified spatially aware embedding space, and then uses the embeddings to infer SC-ST mapping and predict pseudo-spatial adjacency between cells in SC data. Semi-simulation and real data experiments verify that the embeddings preserved spatial information and eliminated technical biases between SC and ST data. We apply STEM to human squamous cell carcinoma and hepatic lobule datasets to uncover the localization of rare cell types and reveal cell-type-specific gene expression variation along a spatial axis. STEM is powerful for mapping SC and ST data to build single-cell level spatial transcriptomic landscapes, and can provide mechanistic insights into the spatial heterogeneity and microenvironments of tissues.

[1] MOE Key Laboratory of Bioinformatics and Bioinformatics Division, BNRIST, Department of Automation, Tsinghua University, Beijing 100084, China.
[2] School of Life Sciences and School of Medicine, Center for Synthetic and Systems Biology, Tsinghua University, Beijing 100084, China.
✉email: weilei92@tsinghua.edu.cn; zhangxg@tsinghua.edu.cn

High-resolution single-cell gene expression with spatial information is critical for revealing the mechanisms of cellular organization, embryogenesis, and tumorigenesis[1–5], and could further enable therapeutic developments[6,7]. Recently, spatial transcriptomic (ST) profiling protocols have been rapidly developed and applied to study gene expression in spatial contests of many tissues[8–11]. The most commonly used ST protocol aggregates multiple cells into one spot, providing in-situ gene expressions and spatial coordinates in a limited resolution[12]. On the contrary, single-cell RNA-sequencing (SC) data provide high-throughput gene expression profiles at single-cell resolution. They have advantages in inferring cellular identity, cell states, and trajectories of diverse cell types[13–16], but lack spatial information.

It is desirable to computationally integrate SC and ST data to retain the advantages from both sides to facilitate comprehensive studies of spatial heterogeneities and variations of transcriptomics[17]. The current widely applied integration method is deconvolution, which employs SC or bulk RNA-seq data as references to estimate the cell type proportion of spots in ST data[18–22]. In deconvolution, a spot is regarded as a mixture of a fixed number of cell types, where the cell type number is decided from prior knowledge or algorithms. Such representation cannot link the SC and ST at the single cell level and limits the potential to flexibly discover sub-clusters or continuous gene expression spatial variations within a cell type. Also, ST data are relatively less available compared with SC data. Deconvolution can only transfer the cell-type information in SC data to ST data, but cannot transfer the spatial information to SC data. This makes it hard to build the spatial single-cell transcription landscape based on the massive SC data. Methods that transfer the spatial information in ST data to SC data are needed. Such a single-cell level spatial landscape would greatly help investigate the spatial proximity of different cell types and spatial variations of gene expression for one cell type.

The core of building a single-cell level spatial landscape is to establish SC-ST mapping and SC-SC spatial associations. A few methods[23–27] and a web server[28] have been proposed. One basic idea is to transfer spatial information based on gene expression similarity. Some data integration methods including Seurat[29] are not designed for this task, but they can construct SC-ST integrated graphs for transferring the spatial coordinates from ST to SC. Tangram[23] learns a mapping matrix to convert SC to ST, and the matrix is optimized by minimizing the cosine similarity between the converted and ground truth ST gene expression profile. Some other methods explicitly consider spatial information. Spaotsc[24] uses the optimal transport theory with spatial constraints to learn the SC-ST mapping matrix. CellTrek[25] uses a multivariate random forest model to map cells to spatial locations. scSpace[26] uses a multi-layer perceptron (MLP) to predict the absolute spatial coordinates by gene expression data.

However, the existing methods didn't simultaneously satisfy several factors that should be considered for mapping the SC data into ST. Firstly, it is essential to align the data between SC and ST datasets. We need to address domain gaps such as batch effects and technical biases to ensure the accuracy of the results. Secondly, gene expression profiles contain rich information on cell identity and state, and we need to filter the information on gene expression data to extract the part that is associated with spatial adjacency. Lastly, interpretability is critical to uncover the mechanisms governing tissue spatial organization. We need to identify the genes that determine the spatial location of individual cells.

To overcome these challenges, we propose STEM, a deep transfer learning model that learns SpaTially-aware EMbeddings of both SC and ST data for SC-ST and SC-SC spatial association

inference. STEM features a shared encoder for SC and ST data to obtain their unified embeddings in the same latent space, and two predictors that simultaneously optimize these embeddings during the training stage. By preserving spatial information and eliminating domain gaps between SC and ST data, the optimized embeddings can be used to infer the SC-ST mapping and the pseudo-SC spatial adjacency. In experiments on both semi-simulation and real data applications, STEM outperforms existing methods in inferring spatial associations and preserving spatial topologies. We identified genes that dominate the spatial distribution of cells by interpreting the trained STEM model with the attribution technique. We used STEM to locate and reveal the spatial proximity of rare cell types in human squamous cell carcinoma (hSCC) data. We used STEM to construct the spatial transcriptomic landscape of hepatic lobules at the single-cell level and identified cell-type-specific gene expression variations along a spatial axis. STEM is a powerful method for revealing detailed and accurate maps of cellular spatial relationships, which can provide mechanistic insights at the single-cell level into spatial transcriptomics studies.

## Results

**STEM: Learning spatially-aware embeddings of ST and SC data.** STEM has an encoder-predictor architecture and represents both ST and SC data as embeddings in a unified space. Figure 1 illustrates the model of the STEM method. To address the issue of unstable and noisy representation of absolute spatial coordinates for cells that are too close or too far apart, we use a normalized spatial adjacency matrix between cells in SC data or spots in ST data as the prediction goal of STEM (Methods). In the training stage, we use the embedding of ST data and SC data to reconstruct two predicted spatial adjacency matrices. The ground truth spatial adjacency matrix is calculated according to the spatial coordinates of ST data via a Gaussian kernel. A cross-entropy loss is calculated on each pair of corresponding rows of the predicted and ground-truth matrixes.

Two predicted spatial adjacency matrices are obtained through two non-parameter predictor modules: the spatial-information extracting module and the domain alignment module. The spatial-information extracting module builds up the ST-ST predicted matrix by computing the correlations between the embeddings of spots in ST data. Each row of the matrix is normalized to 1, and thus each row represents the relative distance of one spot to others. The domain alignment module first eliminates the domain gap between SC and ST embeddings by minimizing the maximum mean discrepancy (MMD)[30], and then constructs an SC-ST mapping matrix and an ST-SC mapping matrix. Both matrices are computed by the correlations between the embeddings of single cells in SC data and spots in ST data, and are normalized in a similar way as the ST-ST predicted matrix. The SC-ST mapping matrix describes the relative distance of one single cell to all spots, and the ST-SC is the reverse. STEM constructs an ST-SC-ST predicted spatial adjacency matrix by multiplying these two mapping matrices.

Through minimizing the loss function during the training procedure, both modules simultaneously optimize encoder parameters to achieve meaningful embeddings of SC and ST data. The spatial-information extracting module encourages the ST embeddings to only contain spatial information, while the domain alignment module encourages the SC embeddings to be similar to ST embeddings and contain reasonable spatial information for building the optimal mapping matrices. Unlike unsupervised dimension reduction algorithms such as autoencoder[31] and VAE[32,33] which condense all information into the latent space, STEM uses spatial adjacency to supervise the

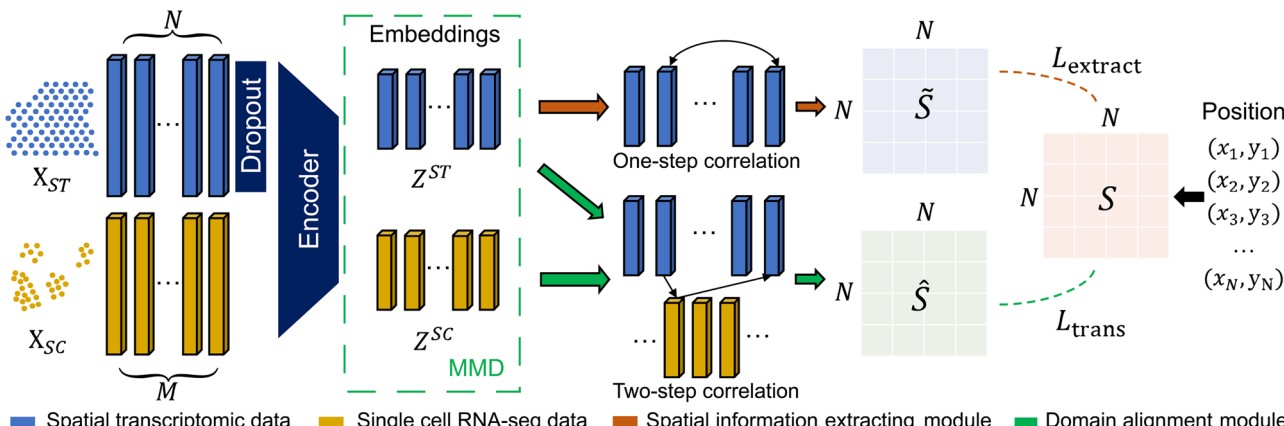

**Fig. 1 Schematic overview of STEM.** Denoting SC and ST gene expression matrices as $X_{ST} \in \mathbb{R}^{N \times H}$ and $X_{SC} \in \mathbb{R}^{M \times H}$, where $N$ and $M$ are spot and cell numbers, and $H$ is the number of genes. To align the sparsity of the $X_{ST}$ with $X_{SC}$, the $X_{ST}$ first passes through an additional dropout layer. Then the processed ST matrix and the SC matrix pass through a shared encoder of STEM to get the corresponding unified embeddings $Z^{ST} \in \mathbb{R}^h$ and $Z^{SC} \in \mathbb{R}^h$ with the same hidden dimension size $h$, respectively. An MMD loss is used to align the distribution of SC and ST embeddings. These embeddings are used to predict the ST-ST spatial adjacency by two modules. The spatial information extracting module uses the correlation between $Z^{ST}$ as the predicted ST-ST adjacency $\tilde{S} \in \mathbb{R}^{N \times N}$. The domain alignment module uses the correlation between $Z^{ST}$ and $Z^{SC}$ to create the cross-domain mapping matrices which are multiplied to generate another ST-ST adjacency $\hat{S} \in \mathbb{R}^{N \times N}$. The reconstruction losses $L_{extract}$ and $L_{trans}$ between the two predicted adjacency and the ground truth adjacency are computed to optimize the STEM encoder. The ground truth spatial adjacency $S \in \mathbb{R}^{N \times N}$ is generated from the spatial coordinate of ST data.

embeddings, helping to extract the spatial information from gene expression and also eliminate the domain gap between SC and ST data.

After training, the optimized ST-SC and SC-ST mapping matrices are used to build the SC-ST spatial adjacency, and the correlation within SC embeddings builds SC-SC spatial adjacency. Additionally, STEM can link the predicted spatial adjacency weights to gene expression. This is because the weight in the predicted spatial adjacency is generated from the embeddings which are encoded from gene expression. By following the spatial adjacency-embedding-gene path, it is feasible to identify the genes that highly contribute to determining the spatial location of each cell. We employ the integrated gradient technique[34] to achieve this. Detailed descriptions of the STEM model and algorithm are provided in the Methods section.

**Semi-simulation experiments showed that STEM achieves accurate spatial mapping at both cell and tissue levels on embryo atlas data.** To benchmark the performance of STEM, we conducted semi-simulation experiments based on the synthetic data generated from the Spatial Mouse Atlas[35] dataset. It is a single-cell resolution spatial transcriptomics dataset. The dataset was produced by seqFISH[36] and contained three distinct mouse embryo slides E1z2, E2z2, and E3z2. On each embryo slide, the single-cell level gene expression profiles were provided and each cell had a spatial coordinate. We treated the gene expression data as pseudo-SC data without the spatial coordinates, and treated the spatial coordinates as the ground truth to test the predictions of methods. We synthesized pseudo-ST data to simulate characteristics of the widely used 10X Visium data: they had a resolution lower than the single cell level and covered only partial cells in the tissue (Fig. 2a). Specifically, we created a grid on the tissue slide and generated pseudo spots at the intersections to cover a portion of single cells in the tissue slide. The gene expression values of each spot were the gene expression summation of all covered cells. The true cell-type proportion of each ST spot was computed based on the cell type annotations of covered single cells.

We applied STEM on the pseudo-SC and pseudo-ST data of each embryo slide to get the unified embeddings and construct

the ST-SC mapping and SC-SC adjacency matrices. We compared STEM with the other five single-cell mapping methods: CellTrek[25], scSpace[26], Seurat[29], Spaotsc[24] and Tangram[23]. As all methods were based on different principles and produced various output forms, we uniformed the outputs of all methods into predicted spatial coordinates, SC-SC adjacency, and SC-ST mapping for evaluation (Methods). We used the same ground truth for evaluating all methods, regardless of how spatial information was used (e.g., normalized or not) in the training process.

We first evaluated whether the absolute spatial location of all single cells can be reconstructed. The reconstruction results of all methods on three embryos were shown in Fig. 2b and Supplementary Figs. 1–3. From these results, STEM was the only method that preserved the original topology structure of all single cells. CellTrek gave a similar shape, but it predicted spatial information for only about 38% of single cells, with the rest discarded by their algorithm. We used the mean absolute error (MAE) between the predicted and ground-truth spatial coordinates to measure the accuracy of predicted coordinates. CellTrek, scSpace, Seurat, and Tangram achieved similar performances while Spaotsc had the highest error (Fig. 2c and Supplementary Fig. 4). STEM consistently achieved the lowest MAE compared to all the other methods.

We then validated the correctness of predicted SC-SC adjacency by hit number which was defined as the number of one cell's true k-nearest neighbors that are successfully predicted (Method). STEM got the highest hit number, about two-fold of the second-best method's performance on all three embryos (Fig. 2c middle and Supplementary Fig. 5). When considering 200 true neighbor cells, STEM identified ~100 correct neighbors on E1z2 data. It is interesting to notice that while for most methods including STEM, a lower MAE corresponded to a higher hit number, CellTrek achieved the low MAE but got the lowest hit number. This might be caused by the cell discarding feature and point repulsion process used in their method. We further separated the single cells into two groups based on whether they were included within spot. We evaluated the Hit number and MAE performance of STEM on these two groups and found that

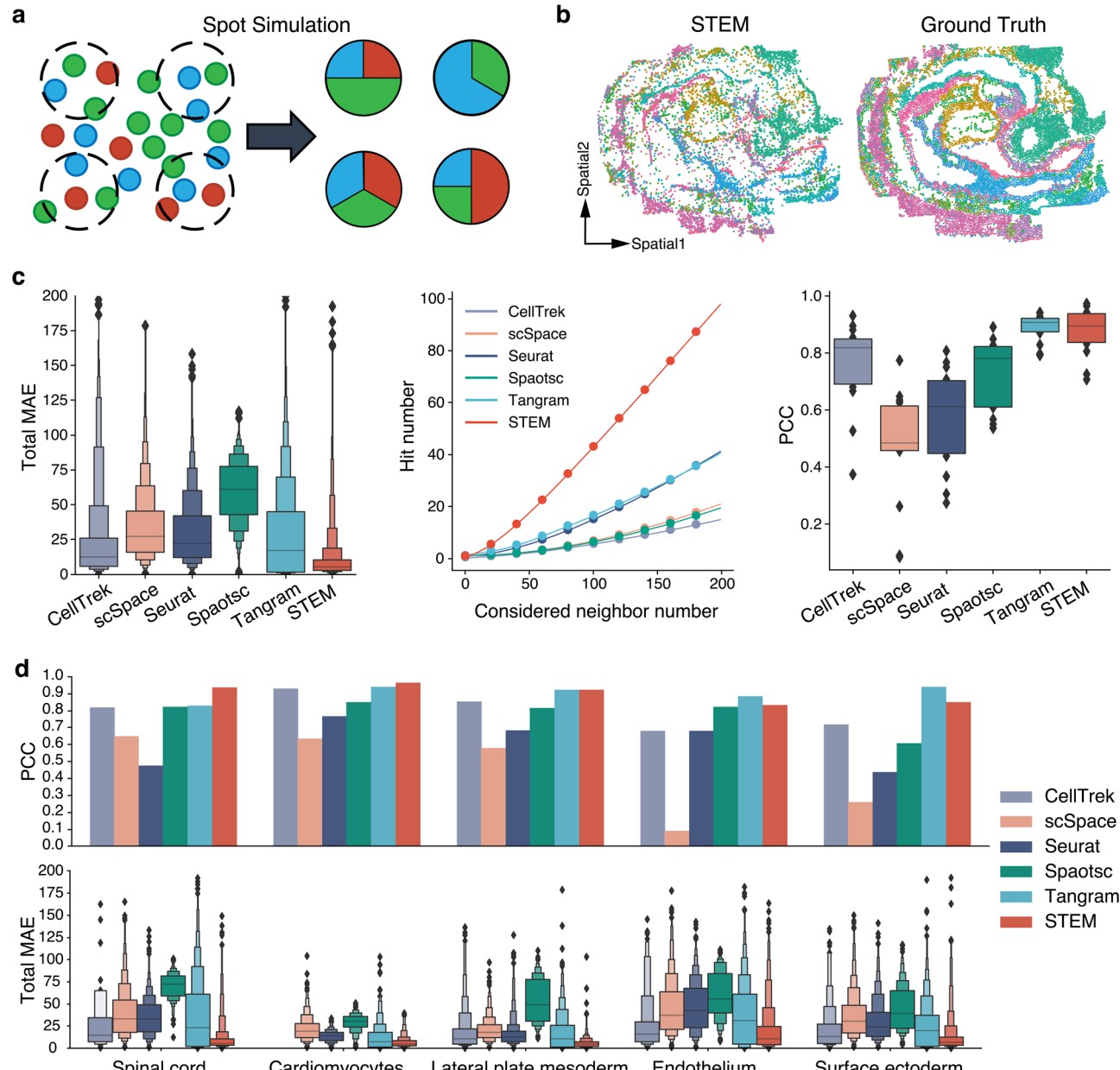

**Fig. 2 The performance evaluation results of different methods on a semi-simulation experiment using mouse embryos. a** An illustration of pseudo-ST data generation. Spots on ST data contain only a fraction of single cells. **b** The reconstructed spatial distribution by STEM versus the ground truth spatial distribution. Colors indicate different cell types or regions. **c** The mean absolute error (MAE), hit number and Pearson correlation coefficient (PCC) performance of different methods on the first mouse embryo data. The lower the MAE, the better. The higher the hit number and PCC, the better. In PCC results, the two edges of box and horizontal bar inside the box represent the interquartile and median of all values, respectively. **d** The PCC and MSE performance of methods on five cell types. These manually selected cell types covered all comparison results (equal, lower and higher) between the PCC of Tangram and STEM. The bar plot shows the PCC between ground truth and predicted spatial distributions of five cell types. In MAE results, we used an enhanced boxplot to show more quintiles. The horizontal bar inside the box represents the median of all values. Each edge of the box represents the half percentiles of the rest data, in other words, splitting the rest data into two halves.

both achieved high hit numbers and low MAEs compared to other methods. The performance on included cells was slightly better than excluded, as expected (Supplementary Fig. 6).

We then evaluated whether the true cell-type spatial distributions can be correctly mapped from single cells to spots. We used the ST-SC mapping matrix to transfer the cell-type annotation from single cells to ST spots and computed the Pearson correlation coefficient (PCC) between the predicted and true distribution of cell types (Methods). As the cell type distribution could be obtained by the deconvolution methods, we also compared the results with Cell2location, a representative method benchmarked previously[21]. STEM and Tangram had comparable average PCCs for all cell types across all three embryos ($0.85 \pm 0.02$ and $0.87 \pm 0.01$, mean ± s.d.), and Cell2location had a similar performance (Fig. 2c and Supplementary Fig. 7). All other methods gave mean PCCs lower than 0.8. We found that the performance is also influenced by the properties of cell types (Supplementary Figs. 8 and 9). For example, cardiomyocytes had obvious aggregation patterns on slides, making it easier to map their spatial distribution by just identifying the cell type from

gene expression. As a result, all methods yielded good results on cardiomyocytes. Conversely, the distributions of hematoendothelial progenitors were scattered in space, necessitating a more detailed subdivision of cell subtypes for mapping cells into their locations. All methods achieved lower PCCs on this cell type. We also ranked the cell types by their contained cell numbers and regarded rank 1–5, 6–10, and other cell types as major, moderate, and rare cell type groups, respectively. As shown in Supplementary Fig. 10, PCCs of all methods' results dropped from major to rare cell types. Tangram and STEM could achieve a median PCC above 0.7 even on the rare cell type.

By comparing the MAE and PCC performance, we observed that Tangram and Spaotsc could map cells to spots with high gene expression similarity, but could not guarantee that these cells were the spatial neighbors of that spot. We further verified this on the five cell types shown in Fig. 2d, Spaotsc reached the average PCC values of all methods, and Tangram had comparable PCC values with STEM. But with the MAE metric, Spaotsc performed worse than the average, while Tangram was also inferior to STEM. These results demonstrated that spatial information is not an explicit signal in gene expression, while STEM achieved more accurate single-cell spatial reconstruction by embedding spatial information.

We conducted experiments to further examine whether STEM's performance was robust across different scenarios (Methods). We constructed different ground truth spatial ST adjacency matrices by changing the parameters and types of the spatial kernel. We found that except for the extremely small values of parameters, in most cases the STEM performance was not sensitive to kernel settings (Supplementary Figs. 11 and 12). When adding noise with different levels into the SC data, we found that the increase in noise didn't decrease STEM's performance too much. The hit number of 200 considered neighbors was above 60, and the median PCC of cell type proportions was above 0.7 (Supplementary Fig. 13). We also simulated the pseudo-ST data with more resolutions including 55, 40, 22, and 10 μm, and we found that STEM had better performance when the resolution increased (Supplementary Fig. 14).

Overall, these semi-simulation experiments systematically showed that STEM could reconstruct the spatial landscape of all SC data by transferring information from ST data that cannot reach a single-cell resolution. STEM consistently achieved more accurate single-cell spatial adjacency estimation compared with other methods and was robust under different settings and noise levels.

**STEM builds spatial informative embeddings and identifies spatial dominant genes**. Interpretability is important for using machine learning-based prediction methods to study underlying mechanisms. We interpreted STEM by analyzing the latent embeddings and traced genes that contribute to the spatial information of cells. For instance, we experimented on single cells in the forebrain, tegmentum, midbrain, and hindbrain regions of E1z2 data. These regions were reported to have spatially-driven transcriptional heterogeneity[35]. The latent embeddings of cells obtained from the STEM encoder showed a trend from the forebrain to the hindbrain in the uniform manifold approximation and projection (UMAP) plot (Fig. 3a), suggesting the ability of STEM for extracting spatial manifold from gene expression profiles.

We found a clear hollow structure on the UMAP of STEM embeddings, but the structure was not shown in the UMAP of PCA embeddings of gene expression derived by SCANPY[37] (Fig. 3a). We examined the hollow structure by calculating the

proportion of cells included in high-dimensional spheres centered on the mean embedding but with different radius (Methods). A hollow structure existed if many cells were excluded from spheres with small radius. As shown in Supplementary Fig. 15, when the radius was low, the proportion of cells embedded by STEM was much lower than that embedded by PCA. All the results suggested that STEM can preserve the spatial structure.

By utilizing integrated gradient techniques[34] on the STEM model, we assigned each cell an attribution vector that showed the contribution of each gene in determining the spatial location of this cell (Methods). We identified genes that had a high contribution to determining cell spatial locations in the STEM model as spatial dominant genes (SDGs). Specifically, we focused on the spinal cord region which is spatially distributed along the anterior-posterior axis. We formulated a spatial trajectory on it and computed pseudo time score for each single cell (Method). We divided the trajectory into 11 segments based on the score (Supplementary Fig. 16) and made the Wilcoxon rank sum test across segments. We identified genes with significantly highly expressed attribution scores (FDR < 0.05) compared to other segments as SDGs. A total of 272 SDGs were identified among 351 genes. The top differentially scored SDGs displayed a clear diagonal pattern in the heatmap (Fig. 3b), indicating that the attribution scores of these top SDGs peaked only in the specific spatial region. We further plotted six SDGs' raw and reconstructed spatial gene expression patterns in Fig. 3c, d. A comparison with their attributions on the heatmap revealed that these genes had high expression values in spatial segments with high attribution scores. We then assessed if the expression patterns of these genes remained consistent in the STEM reconstructed results. We computed cells' estimated pseudo time based on STEM reconstructed spatial locations, and then compared these SDGs expression trends along the reconstructed and ground-truth pseudo time. The Pearson correlation between these two fitted trends was higher than 0.99 (Supplementary Fig. 17), revealing that STEM preserved the genes' expression patterns.

The identified SDGs had potential interests in revealing tissue organization mechanisms, as supported by previous studies. For instance, *Marcks* was reported to be highly expressed in the nervous system and was important for the regulation of embryo development[38]. The Hox genes (*Hoxb5* and *Hoxb1* in our case) were reported to emerge gradually from the posterior aspect of the vertebrate embryo and displayed anteroposterior positional information during tissue generation[39]. There were also some SDGs that did not visually exhibit a strong spatial aggregated expression pattern, such as *Nebl, Bak1, Kmt2d, Suz12*, and *Fgfr2* (Supplementary Fig. 18). We guessed that these genes may help STEM to locate cells in regions with certain cellular states. For instance, *Nebl* is a protein-coding gene involved in the actin-binding and cytoskeletal protein-binding molecular function reported in the Mouse Genome Informatics[40]. The cytoskeletal protein is responsible for many cell functions including cell movements and differentiation. We thus inferred that *Nebl* may help STEM to identify the tissue domain where cell differentiation or movement processes were activated. *Bak1* is a protein-coding gene related to the apoptotic signaling pathway[41] and had been reported to be involved in mouse organogenesis and morphogenesis[42]. We thought this gene may help STEM to identify the local tissue region with an activated apoptotic process. All these results demonstrated that STEM can extract spatial information from genes that do not have easily identifiable spatial patterns.

To compare SDGs with region-specific highly expressed genes, we conducted the Wilcoxon rank sum test on the gene expression matrix to identify differentially expressed genes (DEGs) in

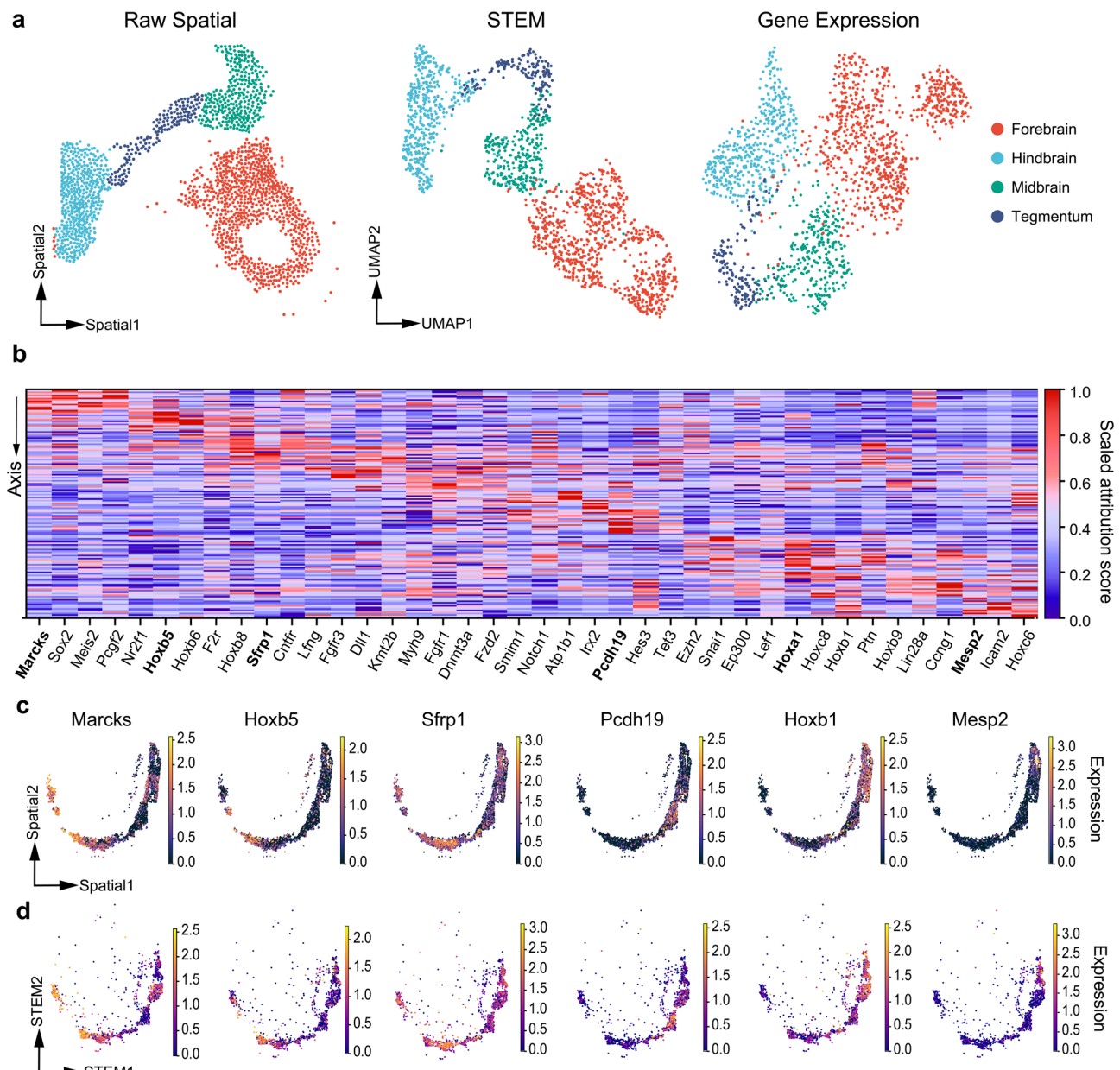

**Fig. 3 Interpretation of the STEM model. a** Raw spatial distribution of cells and UMAP visualization of cell embeddings obtained from STEM and Gene expression. The color indicates the different spatial region annotation. **b** The heatmap shows the attribution score of SDGs along the spinal axis. Each column represents a gene expression vector, with the attribution score scaled from 0 to 1. **c** The ground truth spatial expression patterns of six SDGs. **d** The STEM reconstructed spatial expression patterns of six SDGs. The genes are highly attributed in different regions, corresponding to the bolder name in the heatmap.

segments. As shown in Supplementary Figure 19, while DEGs showed a similar relative rank as SDGs, SDGs had lower FDR values and thus included more genes (272 vs. 218 when FDR < 0.05). For those genes uniquely identified as SDGs, we ranked them by their FDR values and explored the top 10 significant ones. Their attribution profiles had clearer patterns compared with gene expression (Supplementary Fig. 20). These results showed that SDGs were similar but not equivalent to DEGs, and could provide extra information in understanding how genes were related to cells' spatial organization.

These results manifested that the attribution analysis could interpret the STEM model results and made it possible to identify genes that have high contributions for determining cell location, which could provide insights into the spatial formation and evolution of cells in complex normal tissues or tumor microenvironments.

**STEM reconstructs single-cell spatial distribution on the human middle temporal gyrus.** We applied STEM on the well-studied human middle temporal gyrus (MTG) region[43] to further verify its performance in the real case. The SC data we used were sequenced by the SMART-seq protocol and derived from 8 donors between 24 and 66 years old[44]. These SC data lacked the spatial coordinates of cells, but had the dissection information of brain subregions and manually annotated cell types. The ST data we used were at single-cell resolution produced by the in-situ sequencing technique MERFISH[45]. The ST data contained about

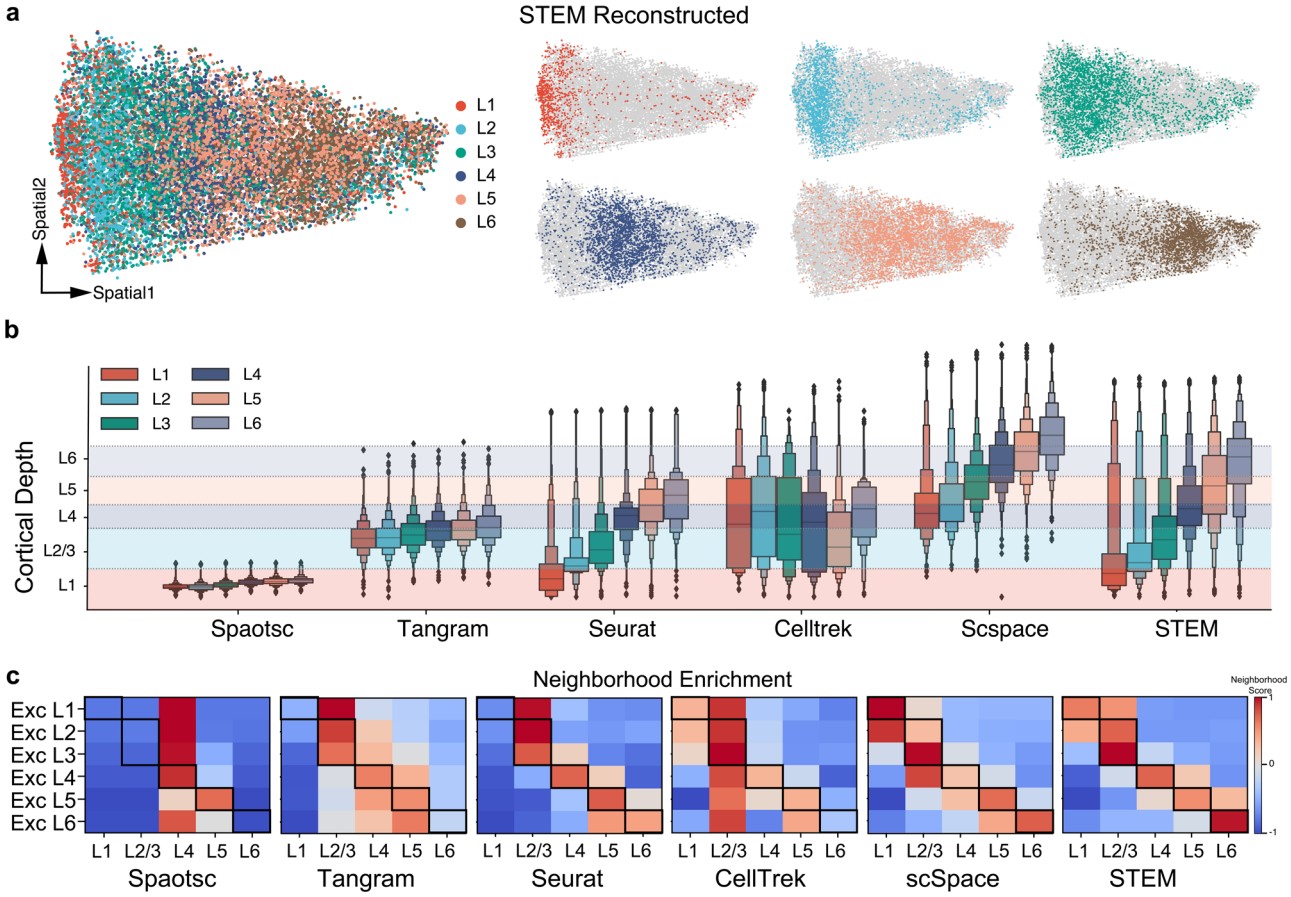

**Fig. 4 Performance evaluation on human MTG using all methods. a** The overall reconstructed spatial distribution of single cells obtained by STEM. The six subplots show the spatial distribution of cells in L1-L6 groups. These groups were determined based on tissue dissection information. **b** Comparison of cortical-depth distribution of cell groups between different methods. The enhanced boxplot in various colors displays the cortical-depth distribution of different single cell groups. The dashed lines indicate the boundaries of layer regions given by ST data. The horizontal bar inside the box represents the median of all values. Each edge of the box represents the half percentiles of the rest data, in other words, splitting the rest data into two halves. **c** Neighborhood enrichment analysis between SC and ST data using all methods. The *x* axis represents regions in the ST data while the *y* axis represents SC excitatory neurons from different dissection layers. The score is row-normalized, and the red color indicates a higher neighborhood score. Bold squares represent areas where high scores are expected.

4000 genes with spatial coordinates and layer segmentation information.

The single-cell spatial distribution reconstructed by STEM closely resembled the human cortex topological structure in the ST data (Fig. 4a), whereas other methods produced more blurry distributions (Supplementary Fig. 21). We further examined the cortical-depth distribution of single cells estimated by all methods across different dissection layer groups. We divided single cells into six layer groups of L1–L6 based on the tissue dissection information. We divided the ST data into L1, L2/3, L4, L5, and L6 subregions using the reference provided in the original study. Ideally, cells from the same dissection layer group should aggregate in the corresponding layer spatial region on the MERFISH tissue. As depicted in Fig. 4b, the cortical-depth distribution of single cells produced by all methods exhibited laminar organization, meaning that cells in the L1 group had the minimal relative depth, while cells in the L6 group had the maximum relative depth. However, all other methods except STEM failed to locate the cells in their corresponding regions. Spaotsc and Tangram compressed all cells into L1 and L2/3 regions. Seurat and CellTrek located cells of L6 group in the shallow region, while scSpace located all cells with an offset to deeper region. Only the depth distribution estimated by STEM fitted well with the true layer regions identified from ST data.

We further validated this result by computing the neighborhood enrichment score[46] between the SC and ST data. We focused on the excitatory neurons in the SC data. As the SC data did not contain any exact spatial information, we compared the results according to the cell annotation in both data to see if cells annotated with the same layer were enriched in spatial. As shown in Fig. 4c, the STEM results had a clear diagonal neighborhood score on the heatmap, indicating the estimated SC spatial distribution of all layers was in accordance with the ST ground truth. Spaotsc mapped all cells around the L4 region. Tangram and Seurat failed to locate the L1 excitatory neurons which is a thin region in human MTG. scSpace mixed the L3 and L4 excitatory neurons. STEM was the only method that recovered the absolute spatial distribution and preserved the spatial topology.

Based on the reconstructed coordinates, we explored whether STEM had the potential to retrieve more genes' spatial patterns that were not captured in ST data. We divided mapped single cells into five regions (L1, L2/3, L4, L5, and L6) according to their reconstructed locations, and found 5 SC unique genes with clear spatial region-specific patterns: *CXCL14, CUX2, RORB, NFIA, APOD*. To verify our finding, we compared results with a ST Visium dataset sequenced on similar human MTG tissue[47]. As shown in Supplementary Figure 22, all five genes exhibited

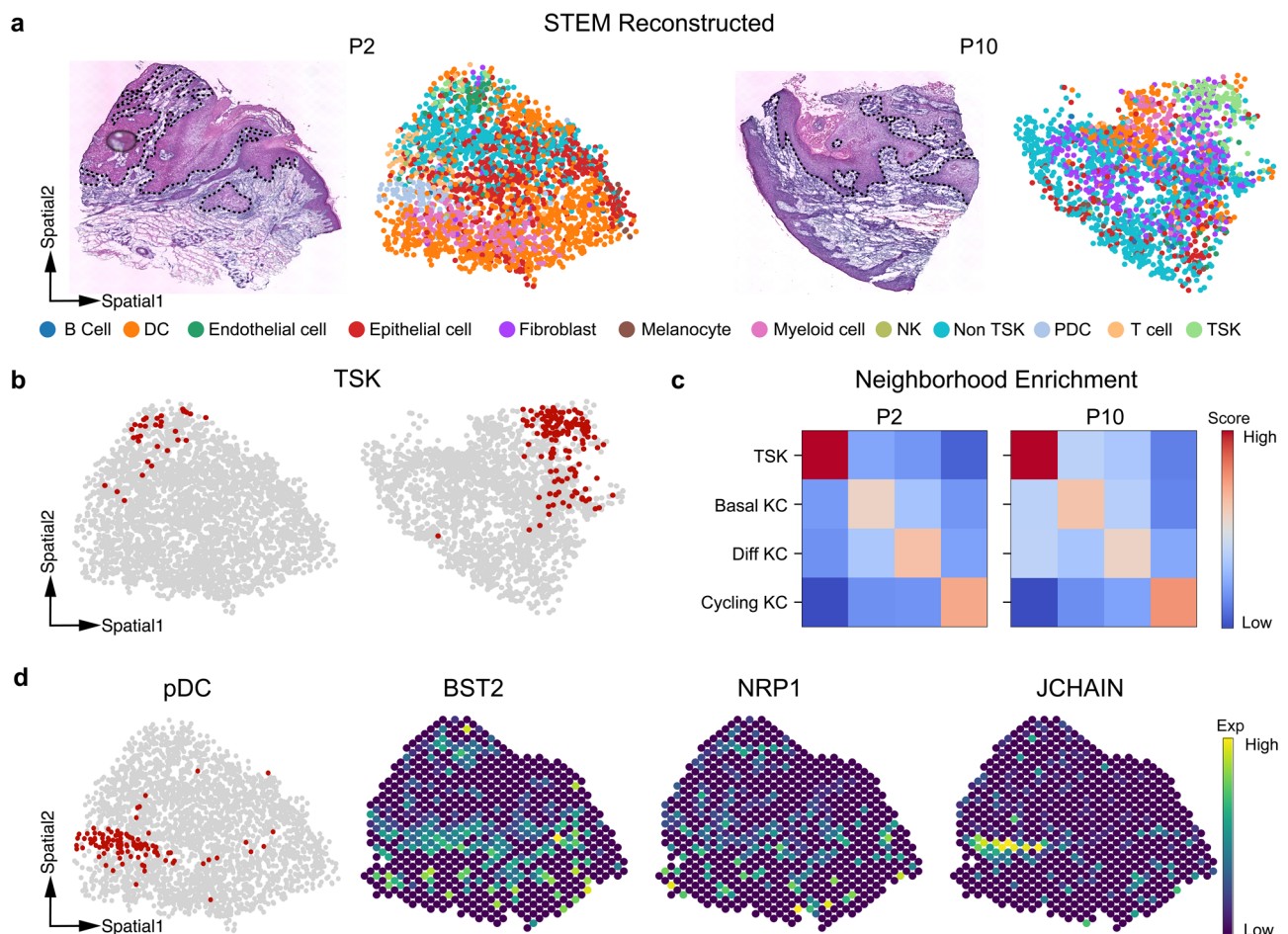

**Fig. 5 STEM results on human squamous cell carcinoma data. a** The HE images and spatial reconstruction results of STEM on patient 2 and patient 10. The black dashed line annotates the tumor-non tumor leading edge observed in the image. The colors of dots represent different cell types. **b** Highlighted spatial distribution of TSK cells on P2 and P10 slides. **c** Neighborhood enrichment analysis on tumor keratinocyte subtypes. The score is row normalized and thus asymmetric. Color in red indicates a higher neighborhood score. **d** The spatial distribution of pDC cells and three spatial expression patterns of corresponding pDC and other immune-related marker genes.

similar expression patterns in the Visium data, and were re-identified as region-specific genes. These results showed that STEM retrieved more genes' spatial patterns from scRNA-seq data.

**STEM locates tumor-specific keratinocytes and immune cells at the single-cell level**. We applied STEM to real datasets to study tumor microenvironments. The characterization of the spatial architecture and arrangement of single cells in the tumor microenvironment is critical for understanding tissue heterogeneity and plasticity[7,48,49]. We applied STEM to a human squamous cell carcinoma (hSCC) dataset[50]. We used the paired SC and ST data from the hSCC tissue of two donors. The SC data were obtained by 10× single-cell sequencing, and a detailed manual cell-type annotation was provided. The ST data were obtained by an early version of the 10X Visium (Spatial Transcriptomics) technique[12].

STEM mapped all the SC data to ST slides accompanied with hematoxylin and eosin (H&E)-stained histological images (Fig. 5a). We verified that the tumor specific keratinocytes (TSKs) were colocalized with endothelial cells at the top tumor leading edge in both donors (Fig. 5b). In the original study, the TSK localization was estimated by scoring each spot with TSK-signature genes which were manually identified from scRNA-seq

data. Compared to this process, STEM achieved similar results but reduced the workload and the potential bias in the manual gene selection process. We further explored the spatial distribution of other keratinocyte (KC) subtypes, including tumor basal, tumor cycling, and tumor differentiating KCs (Supplementary Fig. 23). Neighborhood enrichment analysis showed that TSKs tended to spatially self-aggregate and separate from other KCs, especially far away from the tumor cycling KCs. And tumor differentiating cycling and tumor basal KCs were colocalized (Fig. 5c). This is consistent with the finding in the original study that TSKs and other KCs are distributed in different leading edges. These findings revealed the spatial characteristic of TSKs in the tumor microenvironment.

We then studied the immune cell population in the non-TSK leading edge area, where we observed the predominant spatial positioning of plasmacytoid dendritic cells (pDCs) at the bottom leading edges in donor 2 (Fig. 5d). This cell type was around 4% of the total single cell data. We visually checked the expression levels of pDCs and immune marker genes such as BST2, NRP1, and JCHAIN (obtained from CellMarker[51]) and also quantitatively compared the gene expression levels between spots near pDC cells and other spots (Fig. 5d and Supplementary Fig. 24). The results showed these marker genes were highly expressed in the pDC region (P value < 0.01). Furthermore, the original study reported the activation of IFNs-related signaling pathways in this

region, and STEM allowed us to infer that the localization of pDCs could be the potential driving force of this pathway. This inference was consistent with previous studies showing that pDCs secrete high amounts of type 1 interferon[52]. Our results and analysis illustrated that the topology preserved results given by STEM enabled the joint analysis with the HE image, and STEM streamlined the determination of the spatial location of low proportions cell types without the need for manual identification of signature genes.

**STEM reveals cell-type transcriptomic variations along the liver zonation axis**. We applied STEM to mouse liver data to characterize the cell-type-specific transcriptomic spatial variation in hepatic lobules. In the liver, hepatic lobule is a repeated basic anatomical unit[53] that displays a spatial trend from the portal vein (PV) to the central vein (CV) (Fig. 6a). Studying the transcriptomic variation of different cell types along the trend is critical for revealing the mechanism of liver diseases such as cirrhosis and hepatocellular carcinoma[54]. We used scRNA-seq

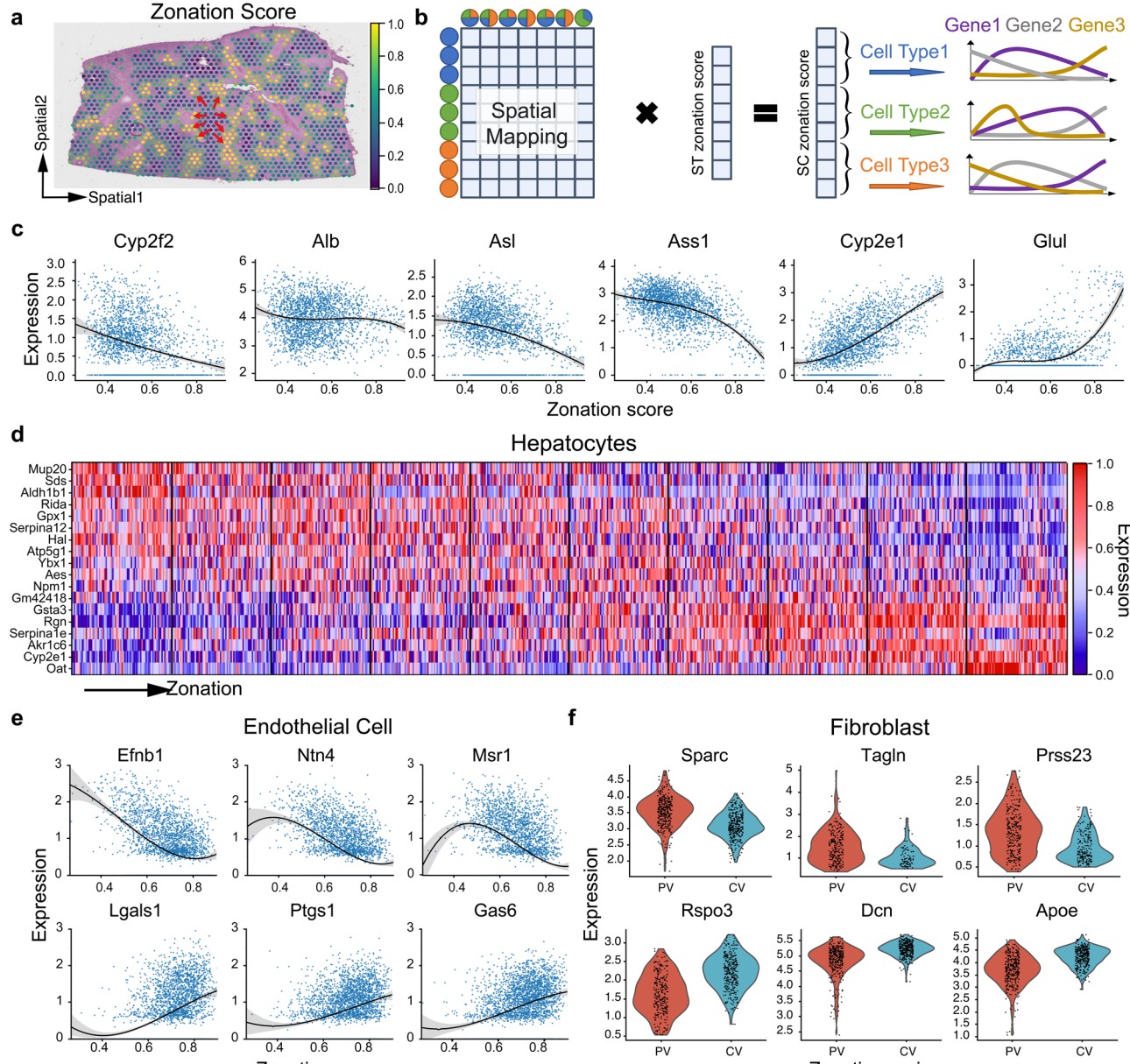

**Fig. 6 Cell-type-specific transcriptomic variation along the liver zonation revealed by STEM. a** Distribution of zonation scores on the ST data. A high score indicates the CV region, while a low score indicates the PV region. Arrow in red indicates the direction from a low (PV) to a high (CV) zonation score region. **b** Illustration of the transfer of zonation scores from ST to SC data. The zonation scores of SC data are obtained by multiplying the SC-ST mapping matrix with the ST zonation score vector. Then cells are grouped into different cell types, and the analysis of cell-type-specific gene variation along the axis can be performed. **c** Expression profiles of six zonation landmark genes along the PV-CV axis. The *x* axis represents the zonation score, and the y-axis represents the gene's raw count expression level. Each curve was obtained by fitting the polynomial function of degree 3 on the corresponding expression value. **d** Heatmap of the top significantly differentially expressed genes along the PV-CV axis. Gene expression values are scaled, with red indicating high expression and blue indicating low expression. **e** Expression profiles of six endothelial cell-specific marker genes along the PV-CV axis. The top and bottom genes are highly expressed in the PV and CV regions, respectively. The shading shows the 95 confidence interval. **f** Violin plots of fibroblast-specific marker genes identified by STEM. The top panel shows the PV marker gene, while the bottom panel shows the CV marker gene.

and ST data from a liver cell atlas[55]. Four liver tissue slide datasets of a healthy mouse from 10X Visium technique were used as the ST data. Single cells from all healthy mice were used as the SC data. Each spot in the ST data was manually assigned a zonation score that reflected the scaled distance from the spot to the PV. Using the spatial mapping matrix obtained from STEM, we reordered the cells along the lobular axis from PV to CV, and revealed the spatial variation of cell-type-specific gene expression along this axis (Fig. 6b).

We first studied hepatocytes which were known to have an obvious spatial variation of gene expression along the PV-CV axis[54,56]. Six previously reported marker genes[57] *Cyp2f2*, *Alb*, *Asl*, *Ass1*, *Cyp2e1*, and *Glul* were highly expressed in the PV, middle, and CV region of the hepatic lobule, as shown in Fig. 6c. By further dividing hepatocytes into 10 spatial subregions along the zonation axis, 819 statistically significant regionally highly expressed genes (FDR < 0.05) were identified. For instance, Gene *Mup20*, *Sds* and *Aldh1b1* were highly expressed near the PV region. Gene *Ybx1*, *Aes* and *Npm1* were highly expressed in the middle region of the trajectory. Gene *Serpinale*, *Akr1c6* and *Oat* were highly expressed near the CV region (Fig. 6d). We validated these region marker genes with a recently published work iSpatial[58]. They gave eight in-situ sequencing verified PV and CV enriched genes, and we found 7 genes were also identified as the significant DEGs of the corresponding near PV and CV subregion (FDR < 0.05, Supplementary Table 1). Furthermore, they computationally identified 10 genes each enriched in PV and CV regions, and we found these genes were also identified as the significant DEGs in the near PV and CV clusters, as shown in Supplementary Tables 2 and 3. These results demonstrated STEM could potentially provide a more comprehensive set of marker genes to reveal the spatially varied transcriptomic properties of hepatocytes.

We conducted a similar analysis on endothelial cells which had also been reported to have spatial variations in a recent study[55]. Two genes *Gja5* and *Adgrg6* were identified as the marker of endothelial cells at the PV region (Supplementary Fig. 25), which was in accordance with the Molecular Cartography results shown in the previous study[55]. Besides, genes such as *Ntn4*, *Msr1*, and *Efnb1* were highly expressed near PV region, and genes such as *Lgals1*, *Ptgs1*, and *Gas6* were highly expressed near CV region (Fig. 6e).

Then, we studied the spatial variation of gene expression in fibroblasts between the PV and CV regions. Fibroblasts are critical to hepatic fibrogenesis and have attracted interest as a potential therapeutic target[59]. The transcriptomic variation among different functional sub-cell types of fibroblasts had been studied in the original study, but the variation in spatial zonation was not investigated. Fibroblasts were categorized into three groups based on their zonation score: those near the PV region, those in the middle, and those near the CV region. The differential analysis was performed between cells near PV and CV regions. 126 and 105 genes significantly highly expressed (FDR < 0.05) of the two regions were found, respectively. Among them, genes *Sparc*, *Tagln,* and *Press23* were the most significantly highly expressed genes near the PV region. *Rspo3*, *DCN* and *Apoe* were the most significantly highly expressed genes near the CV region (Fig. 6f). These genes found by STEM could further help to guide gene selection for spatial in-situ sequencing experiments.

We further performed CellTrek and Tangram, the second and third performed methods in the semi-simulation experiments, on the hSCC and mouse liver dataset, respectively. As shown in Supplementary Fig. 26, for the hSCC data, Tangram was unable to reconstruct the spatial pattern, hindering its application of joint analysis with tissue images. CellTrek reconstructed a coarse morphology in the hSCC dataset but introduced artifacts. For the

mouse liver data, Tangram recovered similar gene expression patterns along the PV-CV axis. The PV and CV marker genes' expression pattern reconstructed by CellTrek was not obvious. These results demonstrated the unique value of STEM for assigning cells with spatial information to study the transcriptomic variation of interested cell types along the tissue anatomic or functional axis.

## Discussion

Revealing the spatial variation of gene expression in tissues and the spatial heterogeneity of cellular transcriptional signatures at the single-cell level is vital to understanding the functional organization of tissues and the underlying mechanisms of various diseases. Due to limitations in current spatial and single-cell transcriptomics techniques, computational methods for constructing spatial gene expression landscapes at the single-cell level are in need. The key is to infer the SC–SC spatial adjacency and SC-ST mapping by learning gene-spatial relations from the ST data. We propose STEM for this purpose. It learns spatially-aware embeddings of transcriptomic data via transfer learning. The learned embeddings support inferring spatial adjacency between spots in ST data and cells in SC data, as well as between cells in SC data.

STEM offers unique advantages for the integrated analysis of SC and ST data. Compared to most spot deconvolution algorithms requiring prior knowledge of cell types, STEM overcomes the limitations of fixed selection of cell-type numbers and does not use any additional metadata information for learning spatial information of all single cells, allowing for the description of cells' continuous status and the exploration of spatial distribution at different levels. Compared to other spatial mapping algorithms, STEM provides more accurate spatial reconstructions that are consistent with referenced spatial topology as shown in our semi-simulation and biological verification experiments. This enables joint analysis of the location of single cells with referenced tissue images, such as colocalizing TSK cells with the tumor leading edge. STEM also supports the analysis of transcriptomic variation within a specific cell type along the spatial axis. Using the single-cell level spatial landscape of liver tissue reconstructed by STEM, we identified gene expression changes in hepatocytes, endothelial cells, and fibroblasts. Upon the model, by following the spatial adjacency-embedding-gene pathfinding, we highlight the spatially dominant genes (SDGs) in the STEM model. These SDGs can be used for model interpretation and facilitate the discovery of healthy or diseased tissue organization mechanisms. Furthermore, identifying SDGs of cancer cells or disease-related cell types could also be used to provide insights into potential drug targets.

STEM can receive input data of various conditions. The analysis on the MTG and hSCC datasets showed the ability of STEM to integrate SC data with different types (in-situ or sequenced based) of ST data. The mouse liver dataset demonstrated that STEM could produce convincing results when ST and SC data were not from the same donor but from the same healthy region with similar cell type proportions. In the future, we will try to apply STEM to SC and ST data from different donors in the same disease state.

Spatial transcriptomics has become a valuable technique in investigating the biological process in different tissues, and has generated a wealth of data[60]. In the future, it is anticipated that more tissues will be characterized using both SC and ST data. Integrating these two data can provide a comprehensive understanding of cell interactions and spatial niches at single-cell resolution. We expect that STEM will be a valuable method for reconstructing single-cell spatial transcriptomic landscapes and enhancing the understanding of spatial cellular heterogeneity.

## Methods

**Spatial adjacency matrix**. STEM takes spatial gene and single-cell expression as input, and predicts the ST-ST spatial adjacency matrix at the training stage. The corresponding ground truth spatial adjacency matrix is converted from spatial coordinates. As for the input, the spatial gene expression data $X^{ST}$ is a $N \times H$ matrix, where $N$ is the number of spots, $H$ is the number of highly variable genes identified from the standard SCANPY workflow. The single-cell gene expression data $X^{SC}$ is a $M \times H$ matrix, where $M$ is the number of cells. The gene is aligned with ST data to make the unified input for STEM. The gene expression value in these matrices is normalized: The unique molecular identifier counts for each gene is divided by the total counts across all genes, and then multiplied by 10,000 and transformed into a log scale.

Giving the ground truth ST data spatial coordinates $Y^{ST} \in \mathbb{R}^{N \times 2}$, we convert the absolute coordinate values into a normalized spatial adjacency matrix $S$. Each element $S_{ij}$ in the matrix represents pairwise spatial association strength and is computed by the Gaussian kernel between coordinate $Y_i^{ST}$ of spot $i$ and $Y_j^{ST}$ of spot $j$. The specific form of Gaussian kernel function is:

$$\phi\left(Y_i^{ST}, Y_j^{ST}\right) = \frac{1}{\sqrt{2\pi}\sigma} \exp\left(-\frac{\left(Y_i^{ST} - Y_j^{ST}\right)^2}{2\sigma^2}\right)$$

$$S_{ij} := \frac{\phi\left(Y_i^{ST}, Y_j^{ST}\right)}{\sum_k S_{ik}}$$

where $\sigma$ is the standard deviation which controls the width of the Gaussian bell, $Y_i^{ST}$ and $Y_j^{ST}$ are the coordinates of two ST spots. Then L1 normalization is applied over columns in $S$ to let the sum of values in each row be 1, and thus each row is the normalized distance from spot $i$ to all spots. From the semi-simulation experiments, we empirically found that STEM achieved the best performance when $\sigma$ is half of the mean nearest neighbor distance of ST spots (Supplementary Figure 11).

**Encoder of STEM**. STEM uses a shared MLP encoder to represent SC and ST gene expression vectors as embeddings in a unified latent space. As a spot in ST data contains more cells than a single cell in SC data, the number of expressed genes in one spot is generally higher than in a single cell, resulting in a higher sparsity of the SC data. To account for this difference in sparsity, we add a dropout layer for ST data before the encoder. The hyperparameter $d$ in the dropout layer which represents the probability of an element to be zeroed is defined as:

$$d = 1 - \frac{\text{Median}(n_{SC})}{\text{Median}(n_{ST})}$$

where $n_{SC}$ and $n_{ST}$ represent the average number of expressed genes in SC and ST data, respectively. Intuitively this dropout layer will set the sparsity of two data in the same level. Then we use a MLP encoder to embed the expression vectors into latent embeddings $Z^{ST} \in \mathbb{R}^h$ and $Z^{SC} \in \mathbb{R}^h$ with the same dimension size $h = 128$.

**Predictor of STEM**. Based on the embeddings obtained from the encoder, STEM reconstructs the spatial adjacency relationship of ST data in two ways, corresponding to the two predictor parts in the model. The first predictor utilizes only ST embeddings to reconstruct spatial relationships, while the second predictor utilizes both SC and ST embeddings.

In the spatial information extracting module, the spatial information extracting module uses ST embeddings $Z^{ST}$ to construct the predicted ST spatial adjacency matrix $\widetilde{S} \in \mathbb{R}^{N \times N}$:

$$\widetilde{S}_{ij} := \text{softmax}_{col}\langle Z_i^{ST}, Z_j^{ST}\rangle = \frac{\exp\left(\langle Z_i^{ST}, Z_j^{ST}\rangle\right)}{\sum_k \exp\left(\langle Z_i^{ST}, Z_k^{ST}\rangle\right)}$$

where $\langle \cdot, \cdot \rangle$ represents the inner products, $softmax_{col}$ denotes the softmaxing operation over columns.

Then STEM uses the cross entropy $H$ as the loss function between the ground truth and predicted ST spatial adjacency matrix $\widetilde{S}$ and $S$. Specifically, the row vectors $\widetilde{S}_i$ and $S_i$ represents the predicted and ground truth normalized distance from spot $i$ to all spots, respectively. The cross entropy loss is applied to these row vectors and can be described as:

$$H_i\left(\widetilde{S}_i, S_i\right) = -\sum_{j=1}^{N} \log \widetilde{S}_{ij} \times S_{ij}$$

The total cross entropy loss is defined as the mean of all rows' loss:

$$L_{\text{extract}} = H\left(\widetilde{S}, S\right) = \frac{\sum_{i=1}^{N} H_i\left(\widetilde{S}_i, S_i\right)}{N}$$

In the domain alignment module, STEM first reduces the mean distance between ST and SC embeddings, and then uses these embeddings to estimate the SC-ST and ST-SC mapping matrices. These two mapping matrices are multiplied together to construct another ST spatial adjacency matrix $\hat{S} \in \mathbb{R}^{N \times N}$. By directly optimizing $\hat{S}$, the optimal SC-ST and ST-SC mapping relationship can be found. A similar idea is proposed in Haeusser's work[61,62]. We extend its applicability from classification to relation construction and fully utilize the cross-domain association matrix as the SC-ST and ST-SC mapping matrix.

Specifically, the STEM introduces the Maximum Mean Discrepancy loss to reduce the mean distance of ST and SC embeddings:

$$L_{\text{MMD}} = \text{MMD}\left(Z_b^{SC}, Z_b^{ST}\right)$$

where $Z_b^{SC}$ and $Z_b^{ST}$ are SC and ST embeddings in a mini-batch. The inner-product is used to measure the similarity between ST and SC embeddings: $B_{ij} := \langle Z_i^{SC}, Z_j^{ST}\rangle$. The mapping matrix $C \in \mathbb{R}^{M \times N}$ from SC to ST is computed by softmaxing similarity matrix $B$ over columns:

$$C_{ij} := \text{softmax}_{col} B_{ij} = \frac{\exp\left(B_{ij}\right)}{\sum_k \exp\left(B_{ik}\right)}$$

Similarly, the mapping matrix from ST to SC $\hat{C} \in \mathbb{R}^{N \times M}$ is computed in the same way by replacing $B$ with $B^T$. Then the two-step spatial adjacency matrix $\hat{S}$ is the multiply results of two mapping matrices:

$$\hat{S} = \hat{C} \cdot C \text{ where } \hat{S}_{ij} := \left(\hat{C} \cdot C\right)_{ij} = \sum_k \hat{C}_{ik} C_{kj}$$

The cross entropy $H$ between the ground truth and this two-step ST spatial adjacency matrix is used as the loss function:

$$L_{\text{trans}} = H\left(\hat{S}, S\right)$$

The total loss of STEM consists of three parts:

$$L = L_{\text{extract}} + \alpha \times L_{\text{MMD}} + \beta \times L_{\text{trans}}$$

The hyperparameters $\alpha$ and $\beta$ are the weight of $L_{MMD}$ and the transition loss $L_{trans}$ in the domain alignment module,

respectively. It should be noticed that one preliminary requirement of successful transferring is that the ST embeddings do encode the spatial information, which is optimized by the $L_{extract}$ loss during training. In addition, the $L_{trans}$ is more important than $L_{MMD}$, because $L_{trans}$ encourages the inner product between ST and SC embeddings could be used to reconstruct the spatial adjacency matrix, and $L_{MMD}$ only encourages minimizing the mean distance between SC and ST embeddings, with no constraint on the spatial relation. We set $\beta$ as a dynamic value, increasing from 0 to 1 during the training. This design encouraged STEM first to focus on reconstructing the spatial adjacency within ST and then learning the spatial mapping between ST and SC.

As for setting the value of $\alpha$, we examined the STEM performance from $\alpha = 0$ to $\alpha = 1$. We used the hit number under the considered hit number of 25 as the evaluation metric. We found that STEM achieved an improved performance (Hit number from 6.3 to 6.6) when introducing the MMD loss ($\alpha > 0$), and the performance (Hit number from 6.6 to 6.7) was not very sensitive to the change of nonzero $\alpha$ values (Supplementary Table 4). So we set the alpha = 0.5 as the default value.

After training, STEM gives multiple results including the cell type deconvolution results $T^{ST} \in \mathbb{R}^{N \times D}$, pseudo spatial coordinates of single cells $\hat{Y}^{SC} \in \mathbb{R}^{M \times 2}$ and the SC spatial adjacency matrix $S^{SC} \in \mathbb{R}^{M \times M}$. STEM uses the ST-SC mapping matrix $\hat{C}$ for deconvoluting the cell type proportion in each spot. Given a cell type indicator matrix $T^{SC} \in \mathbb{R}^{M \times D}$, where D is the number of cell types. Each row in $T^{SC}$ is a one-hot vector, and the nonzero index is the corresponding cell type. The deconvolution result $\hat{T}^{ST} \in \mathbb{R}^{N \times D}$ is given as:

$$\hat{T}^{ST} = \hat{C} \times T^{SC}$$

The pseudo spatial coordinates of single cells are retrieved by computing:

$$\hat{Y}^{SC} = C \times Y^{ST}$$

And the reconstructed spatial adjacency matrix is the inner-product results of SC embeddings:

$$\hat{S}^{SC}_{ij} := \langle Z^{SC}_i, Z^{SC}_j \rangle$$

**Attribution function.** STEM employs the integrated gradient (IG) technique for each cell to assign attribution of genes to its desired spatial location. The IG technique is based on counterfactual intuition, which considers the absence of the cause as a baseline and compares the baseline with current results. For a computational model, the baseline absence of the cause is modeled as a zero input vector. Specifically in STEM, let $X_i$ be the input gene expression of cell $i$, $C_{i,m}$ is the max value in matrix $C$'s $i$ th column, and $X' = [0, 0, \ldots, 0]$ is the baseline vector. Given this three information, the attribution $W_{ij}$ of gene $j$ in cell $i$ is computed as the integrated gradient along the path from baseline $X'$ to the input $X_i$:

$$W_{ij} := IG\left(x_j, x'_j, C_{i,m}\right) = IG\left(x_j, 0, F(x_j)\right) = x_j \times \int_{\alpha=1}^{1} \frac{\partial F(\alpha \times x_i)}{\partial x_i}$$

As the maximum value $C_{i,m}$ indicates that cell $i$ has the maximum probability located around ST spot $m$, the attribution vector $W_i$ reveals the contribution of genes for determining this spatial location. We compute the gene attribution vector of all single cells and get a gene attribution profile $A^{M \times H}$.

**Semi-simulation data generation.** Transcriptomic data with spatial information at the single-cell resolution is required for evaluating the methods' ability of inferring spatial associations. Currently, such data can only be provided by some less popular spatial sequencing technologies which are low-throughput and require complicated operations. We take these experimental single-cell resolution spatial transcriptomic data as SC data with the ground-truth spatial information, and simulate pseudo-ST data by creating a spatial grid on the spatial space of these data.

Specifically, we placed the pseudo-ST spots at the crossing point of the spatial grid. The number of pseudo-ST spots was decided by the topology structure and spatial distribution of the original data. A $30 \times 40$ grid was generated for mouse embryo data as the default setting. The gene expression profile of each spot is obtained by summing the expression of its surrounding single cells. We removed the spots that contained less than three cells. It is noticeable that in real spatial data the tissue cannot be fully covered by spots, so the transcripts of some single cells cannot be captured. We took this into account in the simulation process. The gene expression of each spot aggregated only about 50–70% of the local surrounding single cells. In other words, one-third of single-cell gene expression profiles were not included in the ST data. Then the cell type proportion of each spot is calculated based on its contained single cells' annotation. Pseudo-spots in the simulated ST data have the information of gene expression $X^{ST}$, spatial coordinate $Y^{ST}$ and cell type proportion $P$, where $P$ is a $N \times A$ matrix and $A$ is the number of cell types appeared in the tissue.

**Output unification.** To make the results comparable, we ran all methods with default parameter settings and unified the outputs of different methods into three parts: the reconstructed SC spatial coordinates, the SC spatial adjacency matrix and the SC-ST mapping matrix. The SC spatial coordinates were used for computing MAE. The SC spatial adjacency matrix was used for computing hit number. It was notable that hit number was a rank-based metric and was not sensitive to the kernel function used for building SC spatial adjacency matrix. The SC-ST mapping matrix was used for computing the cell type proportion of spots.

*CellTrek.* We obtained the estimated SC coordinates from Cell-Trek. We set the coordinates of cells discarded by CellTrek as zero. Then we computed the distance between each pair of single cells and got the ranked SC spatial adjacency matrix based on the coordinates. Each row in the adjacency matrix was one cell's spatial proximity to others. The closest cell pair was ranked as one. For getting SC-ST mapping matrix C, given a single cell $i$ and a spot $j$, we first set the value $C_{ij}$ as 1 if the cell was located within the spot in spatial, otherwise 0. Then we normalized the mapping matrix by column, guaranteeing that the sum of cells' mapping weights to one spot was 1.

*scSpace.* We obtained the SC coordinates of all single cells from scSpace. Then we got the SC spatial adjacency matrix and SC-ST mapping matrix via the same procedure used for CellTrek.

*Seurat.* We used the "FindTransferAnchors" to integrate the SC and ST data and got the SC-ST mapping matrix C from "TransferData" function. We got the reconstructed SC spatial coordinates $\hat{Y}^{SC}$ by averaging the coordinates of spots according to the mapping weights:

$$\hat{Y}^{SC} = C^{rn} \times Y^{ST}$$

where $C^{rn}$ was the mapping matrix C with row sum normalized

to 1 and $Y^{ST}$ represented spot coordinates. Based on the reconstructed coordinates, we computed the SC spatial adjacency matrix in the same way illustrated in the above CellTrek paragraph.

*Spaotsc and Tangram.* We obtained the SC-ST mapping matrix $C$ from their outputs. We got the SC spatial adjacency matrix and SC-ST mapping matrix via the same procedure used Seurat.

*STEM.* We obtained the SC and ST embeddings. We got the SC-ST mapping matrix and SC adjacency matrix by computing the inner product of SC-ST and SC-SC embedding pairs. We got the reconstructed SC spatial coordinates by multiplying the mapping matrix with the spot coordinates, as depicted in the "predictor of STEM" section.

**Performance evaluation**. We validate the model performance by using three metrics on the synthetic data. We compute the MAE of distance between the predicted and true spatial coordinates as the error metric. We used the hit number to justify the correctness of predicted SC-SC adjacency. We calculated the PCC between the predicted and true cell type spatial distribution.

The predicted coordinates are computed by multiply the SC-ST mapping matrix with ST spatial coordinate vector. Distance MAE is defined as the mean distance among all predicted and ground truth coordinate pairs:

$$\text{MAE} = \frac{1}{M} \left( \sqrt{\left(Y_0^{SC} - \hat{Y}_0^{SC}\right)^2 + \left(Y_1^{SC} - \hat{Y}_1^{SC}\right)^2} \right)$$

where $Y_0$ and $Y_1$ are the coordinate values in the first and second spatial axis.

Hit number is calculated on the SC spatial adjacency matrix. It is the average number of cell's K-nearest neighbors that can be successfully predicted. It is the function of K and is increased as the K increases. Specifically, for each cell $i$, a set $S_i^g$ contains K ground truth nearest neighbor cells and a set $S_i^p$ contains K predicted truth nearest neighbor cells obtained from the reconstructed SC adjacency. Both two sets have K elements:

$$\left|S_i^g\right| = \left|S_i^p\right| = K$$

The number of cell $i$'s conserved neighbor is:

$$\#\text{ConservedNeighbor}_i(K) = |S_i^g \cap S_i^p|$$

The hit number under K neighbors is the mean value of all cells' conserved neighbor:

$$\text{Hit number}(K) = \frac{1}{M} \sum_{i=1}^{M} \#\text{ConservedNeighbor}_i(K)$$

For cell type $i$, it has a spatial proportion distribution among all ST spots. The ground truth proportion $P_i \in \mathbb{R}^{1 \times N}$ is obtained by recording the annotation of cells belonging to each spot during the semi-simulation generation. As for estimated cell distribution $\hat{P}_i \in \mathbb{R}^{1 \times N}$, we first select the cells (rows) belonging to cell type $i$ in the SC-ST mapping matrix and get a tailored mapping matrix $C^i \in \mathbb{R}^{M_i \times N}$, where $M_i$ is the number of cells belonging to cell type $i$, $N$ is the number of spots. Then, we summarize rows in $C^i$ into a vector $\hat{P}_i \in \mathbb{R}^{1 \times N}$ and regard it as the estimated cell type distribution. The PCC is computed between the estimated cell distribution vector $\hat{P}_i$ and ground truth $P_i$:

$$PCC = \frac{\sum(\hat{P}_i - m_p)(P_i - m_g)}{\sqrt{\sum\left(\hat{P}_i - m_p\right)^2} \sqrt{\sum\left(P_i - m_g\right)^2}}$$

where $m_p$ is the mean of the vector $\hat{P}_i$ and $m_g$ is the mean of the vector $P_i$.

**Constructing ground truth ST spatial adjacency via different kernel settings**. We conducted two experiments on constructing different ST spatial adjacency matrices. First we fixed the Gaussian kernel and varied the parameter $\sigma$ on three semi-simulated embryo data. We used the hit number with considered neighbor numbers ranging from 0 to 25 as the evaluation metric.

Then we constructed the adjacency matrix as the exponential kernel and the KNN kernel. In the exponential kernel, the spatial adjacency value between $Y_i^{ST}$ and $Y_j^{ST}$ was formed as:

$$\phi_e\left(Y_i^{ST}, Y_j^{ST}\right) = \exp\left(-\frac{||Y_i^{ST} - Y_j^{ST}||_2}{l}\right)$$

where $l$ is the scale factor and $|| \cdot ||_2$ is the L2-norm (Euclidean distance). In the experiment, we varied the scale factor $l$ and examined the STEM performance. In the KNN kernel, we defined the spatial adjacency value as 1 if two spots were K neighbors:

$$\phi_{KNN}\left(Y_i^{ST}, Y_j^{ST}\right) = \begin{cases} 1 \, if \, cell \, j \, in \, K \, neighors \\ 0 \, if \, cell \, j \, not \, in \, K \, neighors \end{cases}$$

We varied the number of neighbor K and examined the STEM performance. We plotted and compared the hit number performance with considered neighbor numbers in the range of 0–25.

**Adding different noise levels to semi-simulation data**. For benchmarking STEM on noise SC data, we added the Gaussian noise to the gene expression profiles. Specifically, we simulated the noise gene expression profiles by introducing Gaussian noise into the raw data, characterized by zero mean and gene-specific variances. The noise variance of gene $i$ was calculated as $R \times G_i$, where $G_i$ was the inherent variance of gene $i$. This approach allowed us to manipulate the signal-to-noise ratio via the parameter $R$. To avoid the introduction of negative values due to noise, we established a minimum expression threshold of zero. We generated four noise levels with $R = 0.2, 0.4, 0.6,$ and $0.8$. We plotted and compared the hit number and PCC performance with considered neighbor numbers in the range of 0–25.

**Examining the hollow structure**. We got the cells' embeddings from the STEM model and PCA, respectively. For each type of embedding, we computed the mean embedding and the distance of the nearest neighbor as D. We then regarded the mean embeddings as the center of a high dimensional sphere and monitored how many cells could be included within the sphere as the normalized radius R/D increasing from 0. A hollow structure existed if many cells were not included when the R/D was small.

**Constructing spatial trajectory along the spinal cord**. We fitted a polynomial function $p(x)$ of degree 5 to all cells belonging to the spinal cord region. We regarded the coordinates along the $x$ and $y$ axis as the independent and dependent variables, respectively. And we got the optimal functions by minimizing the squared error:

$$E = \sum_{i=1}^{K} \left|p(x_i) - y_i\right|^2$$

where K was the number of single cells. $(x_i, y_i)$ is the spatial coordinate of cell $i$. After getting the function $p(x)$, we defined the cell at the top right corner as the starting point and set its pseudo time as 0. For other cells, we computed their geodesic distances along the fitted function to the starter cell and used the distances as the pseudo time values.

**Data preprocessing**. We used published data for analysis. The original source data had received ethical approval, and in cases of human data, informed consent from participants.

We obtained the raw count matrices and selected the "$z = 2$" slice for each embryo section. We removed cells with "Low quality" annotation. We used the raw count matrices to generate the pseudo ST data. And then we normalized and log scaled the SC and generated ST data by using the "normalize_total" and "log1p" functions of SCANPY package in Python.

We used the 4000-gene MERFISH data from the donor "H18" as the ST data. We manually separated the MERFISH data into L1-L6 layers according to the Fig. 3 shown in the original paper[43]. We used the exons matrix of SMART-seq data as the SC data. We categorized cells into different layers according to the "brain_region" column in the provided cell metadata. After gene alignment, 3,491 genes were shared across the SC and generated ST data. As for the reference Visium data, we used the data from healthy tissue named "control" from donor "1_1". We aligned their annotations with the MERFISH data by changing "Layer X" to "LX" and merging two pairs "Layer 2" and "Layer 3", "Layer 6" and "White matter" as "L2/3", "L6 and others".

We used single-cell RNA-seq and the Spatial Transcriptomic data of donor 2 and donor 10. We removed single cells with "Multiplet" annotation. To reduce the computation burden, we selected 2000 highly variable genes from the ST data as the common gene set for SC and ST gene alignment. We used the function "nhood_enrichment" in squidpy package to perform the neighborhood enrichment analysis.

We utilized four Visium liver sections from mouse sample 1 as the ST data. We used both CD45− and CD45+ cells (sample "CS88", "CS89", "CS93", "CS97", "CS138" and "CS141") sequenced by scRNA-seq as the SC data. As the number of Endothelial cells and Kupffer cells in the SC data was three times higher than other cell types, we implemented a subsampling strategy with a sampling rate of 0.3. We then identified 2000 highly variable genes in each Visium section and merged them into a union common gene set containing a total of 4866 genes. We used this gene set to align the genes between the SC and ST data.

**Statistics and reproducibility**. All statistical analyses were performed with the SCANPY package in Python. Two independent groups' comparisons were performed using the Wilcoxon rank sum test on the log normalized data, with false discovery rate (FDR) $< 0.05$ considered statistically significant. For finding significant expressed genes across multiple groups or segments, we performed Wilcoxon rank sum test on each group to the union of the rest of the group. We used three mouse embryo tissue slides for the semi-simulation experiments, one human MTG region for the first real application experiment. For the hSCC experiment, we used ST and SC data from two donors, and each donor had three tissue ST replicates. We trained STEM jointly on these three replicates of each donor. For the liver experiment, we used SC and ST data from the same healthy mouse, and the ST data had four tissue slides.

**Reporting summary**. Further information on research design is available in the Nature Portfolio Reporting Summary linked to this article.

## Data availability

The original data used in this paper can be accessed through the following links. The mouse embryo data can be downloaded from https://marionilab.cruk.cam.ac.uk/ SpatialMouseAtlas. The human MTG data can be downloaded from https://doi.org/10. 5061/dryad.x3ffbg7mw, and the SMART-seq data can be downloaded from https:// portal.brain-map.org/atlases-and-data/rnaseq/human-mtg-smart-seq. The referenced human MTG Visium data can be downloaded from GSE220442. The hSCC ST and SC data can be obtained from the GEO database (GSE144240). The mouse liver ST and SC gene expression data can be downloaded from the "Liver Cell Atlas: Mouse StSt" dataset at https://www.livercellatlas.org/download.php. The corresponding spatial coordinates and corresponding H&E images can be obtained from the GEO database (GSE192742). The processed data were deposited in Figshare[63]: https://doi.org/10.6084/m9.figshare. 24452812. The source data for Figs. 2c, 2d, 4b and 6f can be obtained in Supplementary Data 1–4, respectively. All other data are available from the corresponding author (or other sources, as applicable) on reasonable request.

## Code availability

STEM is openly available as a Python package for free academic use. The source code, examples and analysis are accessible at https://github.com/WhirlFirst/STEM or Zenodo[64].

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

## Acknowledgements

This work was partially supported by National Key R&D Program of China (grant 2021YFF1200901), the National Natural Science Foundation of China (grants 62250005, 61721003, and 62103227), and Tsinghua-Fuzhou Institute for Data Technology (TFIDT2021005). This publication is part of the Human Cell Atlas (www.humancellatlas.org/publications/).

## Author contributions

X.Z., M.H., and L.W. conceived the study. M.H., E.L., and Y.W. collected the datasets involved in this article. M.H. and E.L. benchmarked all methods. M.H. designed and implemented the STEM algorithm. Y.C., C.L., S.C., H.G., H.B., and L.W. provided advice on algorithm implementation and simulation experiments. M.H., L.W., Y.W., and J.G. designed the demonstration of biological applications. M.H., E.L., L.W., and X.Z. wrote the manuscript. All authors read and approved the final manuscript.

## Competing interests

The authors declare no competing interests.
