## [Peer Review File · Communications Biology]

This manuscript has been previously reviewed at another Nature Portfolio journal. This document only contains reviewer comments and rebuttal letters for versions considered at Communications Biology.

Reviewers' comments:

Reviewer #1 (Remarks to the Author):

Hao et al. proposed an interesting approach called STEM to map single-cell RNA-seq data onto spatial space. They did benchmark to show STEM outperformed all other 5 existing methods, in tasks like spatial mapping and cell-type deconvolution. I found the described method intriguing and believed that STEM would raise the bar for performance of spatial mapping. However, I would also insist the authors address some issues below.

1. It is not clear to me if the authors also used the normalized spatial adjacency matrix S to train the other 5 methods. It seems to be unfair to report all these evaluation scores if the ground truth is normalized for STEM while not for the other 5 methods.
2. UMAP in Figure 3A is not a justification to show that embedding learned by STEM preserves the hollow structure of Forebrain. Changing parameters in UMAP could likely disrupt the artificial hollow structure in UMAP. The authors need to use another approach to show embedding learned by STEM indeed preserved that spatial hollow structure of forebrain.
3. Identification of spatial dominant genes in Figure 3B is confusing. The author imposed the spinal cord region into different segments and performed wilcoxon rank sum test using the attribution matrix A instead of gene expression matrix. This is an indirect way to identify significant SDGs. Why not directly run the wilcoxon rank sum test with gene expression matrix? Does wilcoxon rank sum test with attribution matrix give the same outcome as wilcoxon rank sum test with gene expression matrix?
4. As STEM maps single cell data of spinal into spatial space, it is worthy of plotting reconstructed spatial expression pattern of those 6 marker genes at single-cell expression level aside the true spatial expression patterns at the seqFISH expression level in Figure 3C. Meanwhile, it is needed to have quantitative report to show that not just each single cell is mapped into spatial space, but also the expression patterns of marker genes of interest are not lost during the mapping.
5. The neighborhood enrichment in Figure 4C lacks a ground truth. It is hard to judge which method gives more authentic representation of spatial neighborhood arrangement. The author will need to set up a ground truth for this comparison.
6. A good application of mapping single-cell RNA-seq into spatial space is that low throughput spatial expression data only profiles a few hundred genes while single-cell RNA-seq data would escalate the low throughput spatial data to the transcriptome level. I did not see the authors provide a good example that STEM helps retrieve more spatial dominant genes from scRNA-seq.

Reviewer #2 (Remarks to the Author):

In this study, Hao et al. developed a deep transfer learning model, STEM, for integrating ST and SC data to achieve single cell level spatial transcriptome. By comparing with other existing methods, STEM shows better performance. The authors also employed STEM to real biology data, which may uncover new biology. Overall, the manuscript is well-written, and the topic is currently trending. Nonetheless, I do have a few concerns.

1. In the fig 2 and fig S1-3, I'm quite surprised at the poor performance exhibited by the other existing methods. At least, according to Tangram paper, it's powerful and can align SC and ST well. Could the authors provide an explanation for this? The authors should provide how they did the benchmark.
2. Cell2location (PMID: 35027729) shares high similarities with this work. It's better to add the cell2location into the benchmark for a comprehensive evaluation.

3. It's commendable that the author apply STEM to real biological scenarios and try to provide new insight. Especially in fig 6, the author provides some genes show expression variations along PV-CV axis. In a recent paper (PMID: 36026447), a similar analysis was conducted. It would be valuable if the authors could compare the results.

Reviewer #3 (Remarks to the Author):

Hao et al. present a method called STEM for mapping single cell and spatial transcriptomics data. STEM applies deep transfer learning to encode both SC and ST data into a joint low-dimensional space, and then uses the embeddings to infer the mapping matrix between SC and ST data. The authors have benchmarked STEM against state-of-the-art methods using both simulations and four real datasets, evaluating the performance in terms of spatial distribution reconstruction and cell type proportion estimation. Overall, STEM appears to be an accurate and efficient method for mapping SC and ST data, and it also aids in downstream analysis. However, the authors should address several critical issues related to the algorithm, simulation design, and conclusions drawn from real data applications. Please refer to my specific comments outlined below.

Major:

1. In general, the method part is clearly written. A notable aspect is the use of the ST data spatial coordinates to construct a spatial adjacency matrix, S , with a fixed Gaussian kernel parameter σ . This was used as the ground truth for training the model. Nonetheless, it's worth noting that the spatial adjacency matrix could differ across various tissue structures.

a. It would be beneficial for the authors to perform sensitivity analysis to see how STEM is robust to different spatial adjacency matrix. For example, how STEM perform when varying σ or when spatial adjacency matrix is constructed from other kernels.

b. The authors used hit number as the evaluation metric to choose the σ in simulations. However, the hit number is related to the number of neighbors considered. More details are needed about the neighbors used in calculating the hit number

c. Selection of hyperparameters α and β : While α has been fixed at 0.5, β has been chosen based on a set of gridded values. A more comprehensive justification or analysis regarding this hyperparameter selection would be invaluable.

2. The manuscript benchmarked STEM with other methods in semi-simulation experiments. However, more details and more comparison analysis are needed to show the robustness and superior performance of STEM.

a. The authors synthesized the pseudo-ST data using a grid-based approach on SC data, and then assessed if each method could accurately reconstruct spatial locations for cells in the SC data. What is the resolution of this pseudo-ST data? Different ST platforms have varied resolutions, and the authors should evaluate performance with changing resolutions of the pseudo-ST data.

b. The authors analyze the SC data that was utilized to produce the ST data. This approach, however, might not truly reflect real-world scenarios where SC data and ST data might come from separate samples and platforms. It would be valuable for the authors to benchmark STEM's performance by introducing various noise levels to the SC data.

c. The authors simulated gene expression for each spot, aggregating only between 50% and 70% of the local surrounding cells. Do the authors then only map the spatial location for the cells that were aggregated in each spot? When the authors apply STEM on the entire scRNA-seq data, how does STEM perform in mapping both the included and excluded single cells? For those cells not included, are they all mapped to their closest original spot (or grid)?

3. In the application to real mouse embryo data, the performance of STEM and other methods differs across cell types. How do various methods perform differently across major, moderate, and rare cell types?

4. The authors claim that STEM can extract spatial information from genes that lack easily identifiable spatial patterns. They further list several examples, such as *Nebl*, *Bak1*, etc. What are the biological interpretations of these genes? How do they relate to tissue structure and domains?

5. The authors did not show the performance of other methods in the hSCC and Visium mouse liver data. How do the other methods perform in these two data?

6. In Figure 5D, the genes *BST2* and *NRP1* do not display the distinct pattern described in the manuscript, which indicates a significant pattern in spots with a high pDC proportion. Could the authors quantify the expression to demonstrate that the expression patterns of these two marker genes align with the cell type proportion distribution?

Minor:

1. In lines 143-149 on Page 5, the authors introduce five single-cell mapping methods. I suggest moving this discussion to the introduction. Furthermore, a more extensive literature review on single-cell mapping methods would be beneficial in the introduction section.

2. There are several places where the notations are unclear or inconsistent:

a. Line 474: is L_{direct} the same as $L_{extract}$ shown in Figure 1?

b. Line 506: T^{SC} should have dimensions $M \times D$ instead of $N \times D$, while T^{ST} should be $N \times D$ instead of $M \times D$, given that M represents the number of cells and N represents the number of spots

c. From line 509 – 513, The notation sometimes includes a " $\hat{}$ " to indicate the estimated value, and sometimes it doesn't. It would be helpful to maintain consistent notation throughout.

3. Several conclusions are inaccurate:

a. Line 166: CellTrek achieved a low MAE but had the lowest hit number. The authors claimed that this might be due to the point repulsion process used in their method. Could it also be related to a large number of discarded cells?

b. Lines 354-355: Should the conclusion be reversed? It seems *Lgals* and others were highly expressed when the zonation score was high, which represents the CV region.

c. The authors suggest that STEM overcomes the limitations of a fixed selection of cell type numbers in traditional spot deconvolution algorithms. However, this might not be true for reference-free deconvolution methods. Furthermore, STEM also uses predefined cell type annotations, which are bound to the fixed number defined in the SC data.

4. More details of the work are needed:

a. Line 221: how did the authors build the spatial trajectory?

b. In Figure 5, why are Basal KC, DiffKC, Cycling KC not present, but only nonTSK shown to represent them all?

c. Line 319, the authors claimed that STEM helps to determine the spatial locations of rare cell types. This claim lacks clarity, as there's no evidence provided that indicates pDC cells are rare

5. Line 352, The endothelial marker genes *Gja5* and *Adgrg6* are missing from Figure 6E

4. Suggestions on visualization: For a clearer understanding of the expression trend along the zonation score line, it would be beneficial for the authors to add several arrows in Figure 6. These arrows could indicate that a transition from a low to a high zonation score represents a shift from the PV region to the CV region.

Point-by-point responses and description of revisions

Reviewer #1 (Remarks to the Author):

Hao et al. proposed an interesting approach called STEM to map single-cell RNA-seq data onto spatial space. They did benchmark to show STEM outperformed all other 5 existing methods, in tasks like spatial mapping and cell-type deconvolution. I found the described method intriguing and believed that STEM would raise the bar for performance of spatial mapping. However, I would also insist the authors address some issues below.

Reply 1.0:

Thank you for the nice summary and positive feedback on our work. We have revised our manuscript to address all the issues. Please find our detailed replies below.

1. It is not clear to me if the authors also used the normalized spatial adjacency matrix S to train the other 5 methods. It seems to be unfair to report all these evaluation scores if the ground truth is normalized for STEM while not for the other 5 methods.

Reply 1.1:

Thank you for your question. The normalized matrix was used only in model training. In the training process, how to use spatial information depends on the loss function designed in different methods. Since the normalized matrix is a feature only designed in STEM, we could not use the normalized spatial adjacency matrix to train the other 5 methods. In the evaluation process, we used the same unnormalized ground truth for all methods and unified the output of all methods to make the results comparable. The comparison is fair for all methods according to their design. We have revised the manuscript to make this point clearer (Line 149-152).

2. UMAP in Figure 3A is not a justification to show that embedding learned by STEM preserves the hollow structure of Forebrain. Changing parameters in UMAP could likely disrupt the artificial hollow structure in UMAP. The authors need to use another approach to show embedding learned by STEM indeed preserved that spatial hollow structure of forebrain.

Reply 1.2:

Thank you for your valuable comments. We expanded our analysis as follows. We focused on the cells located around the hollow structure in spatial and examined whether STEM embeddings

could preserve such structure compared with PCA results shown in Figure 3A UMAP. We got the cells' embeddings from the STEM model and PCA, respectively. For each type of embedding, we computed the mean of all cells' embedding and the mean distance of the nearest neighbor pairs as D . We regarded the mean embedding as the center of a high dimensional sphere and monitored how many cells could be included within the sphere with a certain normalized radius R/D . A hollow structure existed if many cells were not included when the normalized radius was small.

The results are shown in Fig. S15. For embeddings derived from the STEM model, most cells began to be included when R/D was larger than 1. In contrast, for the PCA embedding, many cells were included within the sphere even when R/D was close to zero. In other words, cells in the STEM embedding space exhibited a greater normalized distance from the center, suggesting our methods preserved the hollow structure compared to the PCA. We added the results in the revised manuscript (Line 229-235).

Figure S15. A) The cells around the hollow structure (within the read circle) were selected. B) The cumulative distribution of included cells in the sphere. Y axis: the proportion of included cells. X axis: the normalized radius.

3. Identification of spatial dominant genes in Figure 3B is confusing. The author imposed the spinal cord region into different segments and performed wilcoxon rank sum test using the attribution matrix A instead of gene expression matrix. This is an indirect way to identify significant SDGs. Why not directly run the wilcoxon rank sum test with gene expression matrix? Does wilcoxon rank sum test with attribution matrix give the same outcome as wilcoxon rank sum test with gene expression matrix?

Reply 1.3:

Thank you for the question. In our work, the spatial dominant genes (SDGs) were genes that had a high contribution to determining cell spatial locations in the STEM model. Given one cell's spatial location and its gene expression, performing tests on the attribution matrix would find genes with

high contribution weight to a specific spatial region, conforming to the meaning of SDG. SDGs are similar but not equivalent to the differential expressed genes (DEGs) that have region specific expression values. To show this point, we followed your suggestion for comparing the SDG and DEG. We ran the Wilcoxon rank sum test across different regions (clusters) on the gene expression matrix to identify DEGs. The significant differential expressed genes (DEGs) (with P value < 0.05) found in this way showed a similar relative rank as SDGs (Fig. S19). The SDGs had lower P values and thus more genes were identified as SDG compared with DEG (272 vs. 218). For those genes uniquely identified as SDGs, we ranked them by their FDR values and explored the top 10 significant genes. We found that their attribution profiles had clearer patterns compared with gene expression (Fig. S20), indicating that the genes' contribution to the spatial location was not equal to its expression value and the analysis of the attribution profiles was necessary. We have revised the manuscript to add these results and discussions (Line 270-277).

Figure S19. The scatter plot of genes FDR values. Each subplot corresponds to a cluster in the spinal cord. Each spot is a gene, and the x and y values are the $-\log_{10}$ FDR obtained from the Wilcoxon test results on attribution and gene expression profiles, corresponding to the SDG and DEG, respectively.

Figure S20. Expression profiles of the top 10 unique identified SDG along the Pseudotime. The x-axis represents the pseudotime, and the spots and curves in red and blue colors are the gene's attribution and raw count expression values, respectively. Each curve was obtained by fitting polynomial function of degree 3 on the corresponding expression value.

4. As STEM maps single cell data of spinal into spatial space, it is worthy of plotting reconstructed spatial expression pattern of those 6 marker genes at single-cell expression level aside the true spatial expression patterns at the seqFISH expression level in Figure 3C. Meanwhile, it is needed to have quantitative report to show that not just each single cell is mapped into spatial space, but also the expression patterns of marker genes of interest are not lost during the mapping.

Reply 1.4:

Thank you for your suggestion. We added the reconstructed spatial expression pattern at the single-cell expression level in the revised Figure 3C. For quantitative comparison, we first tagged single cells with estimated spinal cord pseudo time based on their reconstructed spatial locations. Due to directly modeling the gene expressions being noisy, we used the cubic polynomial function to fit the gene expression trends along the reconstructed and ground truth spinal cord pseudo time, respectively. The Pearson correlation between these two fitted functions was higher than 0.99 (Fig. S17), revealing that the mapped single cell still preserved the expression patterns of marker genes.

We have revised the manuscript accordingly (Line 246-253 and Figure 3C).

Figure S17. Expression profiles of six marker genes along the spinal cord pseudotime. The x-axis represents the pseudotime, and the red and blue colors represent the gene's ground truth and reconstructed expression values, respectively. Each curve was obtained by fitting polynomial function of degree 3 on the corresponding expression value.

5. The neighborhood enrichment in Figure 4C lacks a ground truth. It is hard to judge which method gives more authentic representation of spatial neighborhood arrangement. The author will need to set up a ground truth for this comparison.

Reply 1.5:

The single-cell data we used in this experiment do not have exact spatial coordinates and thus we could not get the ground truth numeric neighborhood arrangement matrix. We only had the dissection region information of single-cell data. By using this information, the single cells belonging to different regions could be aligned with regions identified from the spatial MERFISH data, from L1 to L6. So a good criterion we could get was that the square in the dashed line should have a higher enrichment score. And results showed STEM had better performance given this ground truth criteria.

We have added this point to the revised manuscript for readers to better realize this fact (Line 311-313).

6. A good application of mapping single-cell RNA-seq into spatial space is that low throughput spatial expression data only profiles a few hundred genes while single-cell RNA-seq data would escalate the low throughput spatial data to the transcriptome level. I did not see the authors provide a good example that STEM helps retrieve more spatial dominant genes from scRNA-seq.

Reply 1.6:

Thank you for this suggestion. We added the results of retrieving more genes from scRNA-seq data that were not captured in ST data. After reconstructing the single cells' spatial coordinates, we

divided cells into five regions (L1, L2/3, L4, L5, and L6-and-others) according to the space annotation given by MERFISH data. Then we performed Wilcoxon tests between these regions and found 5 region-specific expressed genes with clear spatial patterns: genes *CXCL14*, *CUX2*, *RORB*, *NFIA* and *APOD* were enriched (FDR<0.05) in L1, L2/3, L4, L5 and L6-and-others, respectively.

To verify the results, we used another ST Visium data on a similar human MTG tissue. As shown in Fig. S22, all genes showed a similar expression pattern in the Visium data, and were re-identified as region-specific genes in the Visium data. These results provided the example that STEM retrieved more genes from scRNA-seq data.

We should also mention that these retrieved genes were DEGs but not SDGs, since these genes were not used for training and didn't have the attribution profile. Detailed discussion can be found in Reply 1.3.

We have added these results in our revised manuscript (Line 319-326).

Figure S22. A) The region annotations and five genes' spatial expression patterns on human MTG data. The results in the first row are the STEM reconstructed distribution. The results in the second row are the spatial distribution on the reference Visium data. B) The violin plots of five region specific expressed genes that were newly reconstructed by STEM on human MTG data. The first and second rows correspond to the distribution on STEM reconstructed distribution and reference Visium data, respectively.

Reviewer #2 (Remarks to the Author):

In this study, Hao et al. developed a deep transfer learning model, STEM, for integrating ST and SC data to achieve single cell level spatial transcriptome. By comparing with other existing methods, STEM shows better performance. The authors also employed STEM to real biology data, which may uncover new biology. Overall, the manuscript is well-written, and the topic is currently trending. Nonetheless, I do have a few concerns.

Reply 2.0:

Thank you for the nice summary and positive feedback on our work. We have revised our manuscript to address all your concerns. Please find our detailed replies below.

1. In the fig 2 and fig S1-3, I'm quite surprised at the poor performance exhibited by the other existing methods. At least, according to Tangram paper, it's powerful and can align SC and ST well. Could the authors provide an explanation for this? The authors should provide how they did the benchmark.

Reply 2.1:

Thanks for the question. Tangram directly learned an SC-ST mapping matrix, and the matrix was optimized by minimizing the cosine similarity between the converted and ground truth ST gene expression data, which means Tangram did not introduce any spatial information in the design. As a result, Tangram could map one spot with a group of single cells with high gene expression similarity, but could not guarantee these cells were the spatial neighbor of that spot. This was also shown in our results that Tangram achieved an even better PCC compared to STEM on cell type deconvolution task but had a low hit number (Fig. 2C). A discussion is in the revised manuscript (Line 188-195).

For all methods, we benchmarked them as follows:

1. We executed them with the default parameter settings.
2. As all other methods were based on different principles (summarized in Line 145-150 of the original manuscript) and produced different output forms, we unified their outputs into three parts for the next step evaluation: the reconstructed SC spatial coordinates, the SC spatial adjacency matrix and the SC-ST mapping matrix. The SC spatial coordinates were used for computing MAE. The SC spatial adjacency matrix was used for computing the hit number. It was notable that hit number was a rank-based metric and was not sensitive to the kernel function used for building SC spatial adjacency matrix. The SC-ST mapping matrix was used for computing the cell type proportion of spots. For instance, Tangram only provided SC-ST mapping matrix as the original

output, the estimated SC coordinates were obtained by averaging the coordinates of spots according to the weights of the mapping matrix. Then the SC-SC adjacency matrix was obtained by computing the spatial distance within the SC coordinates.

3. The SC coordinates were used to compute the mean absolute error. The SC-SC adjacency was used to compute the hit number. The SC-ST mapping matrix was used for calculating the PCC of cell type proportion in spots. The computation equation and process were in the method section.

We have extended the manuscript for readers to better understand the difference in methods and the process of our benchmarking (Line 51-59,190-193 and 611-640).

2. Cell2location (PMID: 35027729) shares high similarities with this work. It's better to add the cell2location into the benchmark for a comprehensive evaluation.

Reply 2.2:

Thanks for the suggestion. Cell2location is a powerful tool for using SC data to deconvolute the cell type spatial distribution in ST spots, but it cannot do the mapping between SC and ST data since its deconvolution process is based on the cell type signature matrix. We have revised the manuscript to clarify the difference between deconvolution and single-cell mapping (Line 40-44).

To follow your suggestion, we could only evaluate the cell2location performance on the cell type deconvolution task. The results showed that it consistently achieved comparable performance methods in all three embryo data. We also wanted to point out that cell2location has its advantage in processing data from different batches, which is not in the scope of our semi-simulation experiments.

Figure R1. Pearson correlation coefficient (PCC) performance of different methods on three mouse embryo data. The performance of cell2location is the box in brown.

3. It's commendable that the author apply STEM to real biological scenarios and try to provide new insight. Especially in fig 6, the author provides some genes show expression variations along PV-CV axis. In a recent paper (PMID: 36026447), a similar analysis was conducted. It would be valuable if the authors could compare the results.

Reply 2.3:

Thank you for the suggestion. We compared the STEM results with iSpatial (PMID: 36026447) results from two aspects. First, in their work, the ground truth MERFISH data of eight genes enriched in the PV and CV regions were provided. We found that 7/8 genes were also identified as the significant DEGs (P value<0.05 and Log2FC>0) in our reconstructed results (Table S1). Second, they computationally identified 10 genes each enriched in PV and CV regions. We found these genes were also identified as the significant DEGs in the near PV and CV clusters, as shown in Table S2 & S3. We have added these results in the revised manuscript (Line 397-402).

Reviewer #3 (Remarks to the Author):

Hao et al. present a method called STEM for mapping single cell and spatial transcriptomics data. STEM applies deep transfer learning to encode both SC and ST data into a joint low-dimensional space, and then uses the embeddings to infer the mapping matrix between SC and ST data. The authors have benchmarked STEM against state-of-the-art methods using both simulations and four real datasets, evaluating the performance in terms of spatial distribution reconstruction and cell type proportion estimation. Overall, STEM appears to be an accurate and efficient method for mapping SC and ST data, and it also aids in downstream analysis. However, the authors should address several critical issues related to the algorithm, simulation design, and conclusions drawn from real data applications. Please refer to my specific comments outlined below.

Reply 3.0:

Thank you for the nice summary and positive feedback on our work. Following your suggestions, we have systemically conducted more experiments on the algorithm design, semi-simulation data, and downstream analysis.

Major:

1. In general, the method part is clearly written. A notable aspect is the use of the ST data spatial coordinates to construct a spatial adjacency matrix, S , with a fixed Gaussian kernel parameter σ .

This was used as the ground truth for training the model. Nonetheless, it's worth noting that the spatial adjacency matrix could differ across various tissue structures.

Reply 3.1:

Thank you for the positive comments. We have conducted extra experiments on different spatial adjacency matrices and the results are shown below.

a. It would be beneficial for the authors to perform sensitivity analysis to see how STEM is robust to different spatial adjacency matrix. For example, how STEM perform when varying σ or when spatial adjacency matrix is constructed from other kernels.

Reply 3.1.a:

We conducted two new experiments on constructing the spatial adjacency matrix. First, we fixed the Gaussian kernel and varied σ on three semi-simulated embryo data (E1z2, E2z2 and E3z2). We used the hit number with considered neighbor numbers ranging from 0 to 25 as the evaluation metric. As shown in Fig. S11, in general, the hit number is monotonically increased when the considered neighbor number increases. STEM achieved a similar performance in most σ values. We empirically found that STEM achieved the best performance when σ is half of the ST spots nearest neighbor distance ($\sigma=3$ in E1z2 and E2z2 data, $\sigma=7$ in E3z2 data).

Besides using the Gaussian kernel, we then constructed the adjacency matrix by using the exponential kernel and the KNN kernel. In the exponential kernel, the spatial adjacency value between Y_i^{ST} and Y_j^{ST} was formed as:

$$\phi_e(Y_i^{ST}, Y_j^{ST}) = \exp\left(-\frac{\|Y_i^{ST} - Y_j^{ST}\|_2}{l}\right)$$

where l was the scale factor and $\|\cdot\|_2$ was the L2-norm (Euclidean distance). In the experiment, we varied the scale factor l and examined the STEM performance on the E1z2 data. In the KNN kernel, we defined the spatial adjacency value as 1 if two spots were K neighbors, otherwise 0. We varied the number of neighbor K and examined the STEM performance on the E1z2 data. We observed similar conclusions that except for the extreme values of parameters, STEM could achieve a stable performance (Fig. S12).

All these results showed that STEM was not sensitive to the selection of spatial kernels or changing hyperparameters in an adjacent interval. We have extended the previous description and added these results to our revised manuscript (Line 196-200,670-682).

Figure S11. The STEM hit number performance under different values of parameter σ on all three embryo data.

Figure S12. The STEM hit number performance under different kernels. Left: performance under different values of parameter l in the exponential kernel. Right: performance under different values of parameter K in the KNN kernel.

b. The authors used hit number as the evaluation metric to choose the σ in simulations. However, the hit number is related to the number of neighbors considered. More details are needed about the neighbors used in calculating the hit number

Reply 3.1.b:

In our experiment (as shown in Reply 3.1.a), the hit number was monotonically increased when the considered neighbor number increased. Thus comparing the performance under any considered neighbor number would get similar results. We plotted and compared the hit number with the considered neighbor number in the range of 0-25 to choose the σ . We have added these details in the revised manuscript (Line 651-660, 682, 691).

c. Selection of hyperparameters α and β : While α has been fixed at 0.5, β has been chosen based on a set of gridded values. A more comprehensive justification or analysis regarding this hyperparameter selection would be invaluable.

Reply 3.1.c:

Thank you for the question. The hyperparameters α and β are the weight of L_{MMD} and the transition loss L_{trans} in the domain alignment module, respectively. It should be noticed that one preliminary requirement of successful transferring is that the ST embeddings did encode the spatial information, which is optimized by the $L_{extract}$ loss during training. In addition, the L_{trans} is more important than L_{MMD} , because L_{trans} encourages the inner product between ST and SC embeddings could be used to reconstruct the spatial adjacency matrix, and L_{MMD} only encourages minimizing the mean distance between SC and ST embeddings, with no constraint on the spatial relation.

Considering these aspects, we set β as a dynamic value, increasing from 0 to 1 during the training stage. This design encouraged STEM first to focus on the spatial adjacency reconstruction within ST and then the spatial mapping between ST and SC.

As for setting the value of α , we examined the STEM performance from $\alpha=0$ to $\alpha=1$. We used the hit number under the considered hit number of 25 as the evaluation metric. We found the performance was improved when introducing the MMD loss, and the performance was not very sensitive to the change of nonzero α values (Table S4). So we set the $\alpha=0.5$ as the default value.

We added more results and discussion in the revised manuscript (Line 555-568).

Table S4. Hit number performance under 25 considered neighbors by setting different α values

α	0	0.1	0.2	0.3	0.4	0.5	0.6	0.7	0.8	0.9	1
Hit number (25)	6.35	6.58	6.55	6.65	6.72	6.63	6.58	6.67	6.68	6.69	6.70

2. The manuscript benchmarked STEM with other methods in semi-simulation experiments. However, more details and more comparison analysis are needed to show the robustness and superior performance of STEM.

a. The authors synthesized the pseudo-ST data using a grid-based approach on SC data, and then assessed if each method could accurately reconstruct spatial locations for cells in the SC data. What is the resolution of this pseudo-ST data? Different ST platforms have varied resolutions, and the authors should evaluate performance with changing resolutions of the pseudo-ST data.

Reply 3.2.a:

Thanks for the question. We used the same 30x40 grid on three Embryo data for generating pseudo-ST data. The diameter of the ST spot on E1z2 and E2z2 data is about 22um, and the distance between two spots center is about 40um. For E3z2 data, due to the larger scale in the

original spatial data, the diameter of the ST spot is about 60um, and the distance between two spots center is about 100um.

We simulated the pseudo-ST data with different diameters of 10um, 40um, and 55um by adjusting the number of spots in the grid. The minimal and maximum resolution correspond to the current Slide-seq and 10X Visium platform. In each resolution, we remained spots containing partial cells in the tissue slide. We used hit number, MAE, and the PCC of cell types to evaluate the STEM performance. We found that STEM could achieve a hit number higher than 60 under the diameter of 55um, and with the spot diameter decreased STEM achieved better performance.

We have added these new experiments in the revised manuscript (Line 202-204).

Figure S14. The mean absolute error (MAE), hit number and Pearson correlation coefficient (PCC) performance of STEM on different resolution semi-simulated data.

b. The authors analyze the SC data that was utilized to produce the ST data. This approach, however, might not truly reflect real-world scenarios where SC data and ST data might come from separate samples and platforms. It would be valuable for the authors to benchmark STEM's performance by introducing various noise levels to the SC data.

Reply 3.2.b:

Thank you for the suggestion. We conducted new experiments for benchmarking STEM on SC data with different noise levels. Specifically, we simulated the noise gene expression profiles by introducing Gaussian noise into the raw data, characterized by zero mean and gene-specific variances. The noise variance of gene i is calculated as $R \times G_i$, where G_i is the inherent variance of gene i . This approach allowed us to manipulate the signal-to-noise ratio via the parameter R . To avoid the introduction of negative values due to noise, we established a minimum expression threshold of zero. We generated four noise levels with $R=0.2, 0.4, 0.6,$ and 0.8 .

The results showed that adding noise indeed decreased STEM's performance. But in general, the hit number is above 60 when considering 200 ground truth neighbor numbers, and the median PCC of cell type proportions was above 0.7.

We have added these new experiments and results in the revised manuscript (Line 201-202,683-691).

Figure S13. The mean absolute error (MAE), hit number and Pearson correlation coefficient (PCC) performance of STEM on different noise levels (R) semi-simulated data.

c. The authors simulated gene expression for each spot, aggregating only between 50% and 70% of the local surrounding cells. Do the authors then only map the spatial location for the cells that were aggregated in each spot?

When the authors apply STEM on the entire scRNA-seq data, how does STEM perform in mapping both the included and excluded single cells? For those cells not included, are they all mapped to their closest original spot (or grid)?

Reply 3.2.c:

Thanks for the question and suggestion. We mapped all cells including both unaggregated and aggregated cells into the spatial. Our evaluation is based on all these cells.

Following your suggestion, we separated the cells into the excluded and included groups and evaluated the Hit number and MAE. We found that both included and excluded cells achieved a high hit number and low MAE, and the performance of included cells is better than excluded as expected (Fig. S6). Then we computed the proportion of excluded cells that could be mapped to their closet spot. About 10%-20% of excluded cells could find the correct closet spot. By releasing the criteria as considering the K closest spots, which means we define success as the cells could be

mapped into any of the top K closest original spots, we observed an improvement when K is increase, and about 50% of excluded cells could mapped to the right spots when K=4.

We have revised the manuscript and added the new results (Line 169-172).

Figure S6 The first two rows are Total MAE and Hit number performance of two cell groups given by STEM on all three embryo data, respectively. The third row is the proportion of excluded cells were mapped in two the K closest spots. X axis is the K values, and Y axis is the proportion. The “include” and “exclude” groups contain cells that are included or excluded in the spot.

3. In the application to real mouse embryo data, the performance of STEM and other methods differs across cell types. How do various methods perform differently across major, moderate, and rare cell types?

Reply 3.3:

Thank you for the question. We ranked the cell types by their cell number and regarded the rank 1-5, 6-10, and other cell types as the major, moderate, and rare cell types respectively. As shown in Fig. S10, all methods' performance was dropped from major to rare cell types. STEM could achieve a median PCC above 0.7 even on the rare cell type and was the top-ranked method. We observed similar results on all three Embryo data. We have added these analyses in the revised manuscript (Line 184-187).

Figure S10. The PCC results of different methods among major, moderate, and rare cell types on all semi-simulated data. Each row corresponds to a dataset, and each column corresponds to one class of cell type.

4. The authors claim that STEM can extract spatial information from genes that lack easily identifiable spatial patterns. They further list several examples, such as *Nebl*, *Bak1*, etc. What are the biological interpretations of these genes? How do they relate to tissue structure and domains?

Reply 3.4:

Thanks for the question. While we didn't find the direct biological roles of the genes *Nebl* and *Bak1* in embryonic spinal cord development, we hypothesized their potential influences based on existing knowledge as follows.

Nebl is a protein-coding gene involved in the actin-binding and cytoskeletal protein binding molecular function reported in the Mouse Genome Informatics (MGI)¹. The cytoskeletal protein is responsible for many cell functions including cell movements and differentiation. We think it may help STEM to identify the tissue domain where cell differentiation or movement processes are activated.

Bak1 is a protein-coding gene related to the apoptotic signaling pathway² and it has been reported to be involved in mouse organogenesis and morphogenesis³. We think this gene may help STEM to identify the local tissue region existing in the apoptotic process.

We have added a discussion on these genes in the revised manuscript (Line 260-268).

1. Blake, J. A. *et al.* Mouse Genome Database (MGD): Knowledgebase for mouse-human comparative biology. *Nucleic Acids Research* **49**, D981–D987 (2021).
2. Cao, K., Gong, Q., Hong, Y. & Wan, L. *uniPort: a unified computational framework for single-cell data integration with optimal transport*. <http://biorxiv.org/lookup/doi/10.1101/2022.02.14.480323> (2022) doi:10.1101/2022.02.14.480323.
3. Ke, F. F. S. *et al.* Embryogenesis and Adult Life in the Absence of Intrinsic Apoptosis Effectors BAX, BAK, and BOK. *Cell* **173**, 1217-1230.e17 (2018).

5. The authors did not show the performance of other methods in the hSCC and Visium mouse liver data. How do the other methods perform in these two data?

Reply 3.5:

Thanks. We selected CellTrek and Tangram to be tested on the downstream application datasets, since these two methods were shown to achieve the 2nd and 3rd performance in our semi-simulation experiments.

On hSCC data, we plotted the cells by using the reconstructed spatial coordinates. On Visium liver data, we plotted the recovered gene expression along the zonation axis and computed the PCC between the zonation score and recovered gene expression.

The results are shown in Fig. R2-4. The Tangram method failed to reconstruct the spatial pattern in the hSCC data, hindering its application of joint analysis with tissue images. It could recover similar gene expression patterns along the PV-CV axis. The CellTrek method reconstructed a similar coarse morphology in the hSCC dataset but introduced many artifacts. While in the Visium mouse data, it failed to recover the PV and CV marker genes' expression pattern. STEM is the only method that can support the analysis of both datasets.

Figure R2. The spatial reconstructed results of CellTrek and Tangram methods on the hSCC dataset.

Figure R3. Expression profiles of six zonation landmark genes reconstructed by CellTrek along the PV-CV axis.

Figure R4. Expression profiles of six zonation landmark genes reconstructed by Tangram along the PV-CV axis.

6. In Figure 5D, the genes *BST2* and *NRP1* do not display the distinct pattern described in the manuscript, which indicates a significant pattern in spots with a high pDC proportion. Could the

authors quantify the expression to demonstrate that the expression patterns of these two marker genes align with the cell type proportion distribution?

Reply 3.6:

Thank you for the suggestion. In the original manuscript, we said that these two genes were found to be highly expressed in the same region the pDC cell mapped. However, this observation doesn't necessarily correlate with the distribution of cell type proportions due to the presence of dropout and noise in the gene expression data, particularly from the early ST sequencing platform utilized. To quantify the relation between two genes and pDC, we focused on spots in close proximity to the pDC single cells, aiming to determine whether there was indeed a high expression of these two genes in those specific locations. We did the Wilcoxon test on the gene expression of *BST2* and *NRP1* between the near pDC spots and other spots. And these two genes were identified as DEGs and had 1.11 and 0.98 positive log₂ foldchange with P values 1e-3 and 2e-2 on near pDC spots, respectively.

We have added these analyses to the revised manuscript (Line 360-364).

Figure S24. A) Spots located near the pDC single cells were in orange. B) Violin plot of pDC enriched genes *BST2* and *NRP1*. Both two genes were identified as DEGs in the near pDC region.

Minor:

1. In lines 143-149 on Page 5, the authors introduce five single-cell mapping methods. I suggest moving this discussion to the introduction. Furthermore, a more extensive literature review on single-cell mapping methods would be beneficial in the introduction section.

Reply 3.7:

Thank you for your suggestion. We have revised the manuscript accordingly (Line 51-59).

2. There are several places where the notations are unclear or inconsistent:

a. Line 474: is L_{direct} the same as L_{extract} shown in Figure 1?

b. Line 506: T^{SC} should have dimensions $M \times D$ instead of $N \times D$, while T^{ST} should be $N \times D$ instead of $M \times D$, given that M represents the number of cells and N represents the number of spots

c. From line 509 – 513, The notation sometimes includes a " $\hat{}$ " to indicate the estimated value, and sometimes it doesn't. It would be helpful to maintain consistent notation throughout.

Reply 3.8:

Thank you for these detailed corrections and suggestions. We have corrected the bugs, and also double-checked the whole manuscript for other possible bugs.

3. Several conclusions are inaccurate:

a. Line 166: CellTrek achieved a low MAE but had the lowest hit number. The authors claimed that this might be due to the point repulsion process used in their method. Could it also be related to a large number of discarded cells?

Reply 3.9:

Yes, the discarded cell would also prohibit CellTrek from finding the original nearest neighbor. We have revised it. Thanks.

b. Lines 354-355: Should the conclusion be reversed? It seems L_{gals} and others were highly expressed when the zonation score was high, which represents the CV region.

Reply 3.10:

Sorry for the bug and thanks for pinpointing it for us. We corrected it in the revision.

c. The authors suggest that STEM overcomes the limitations of a fixed selection of cell type numbers in traditional spot deconvolution algorithms. However, this might not be true for reference-free deconvolution methods. Furthermore, STEM also uses predefined cell type annotations, which are bound to the fixed number defined in the SC data.

Reply 3.11:

Thank you for pointing out the possible confusion about not including the reference-free deconvolution methods. We have revised the manuscript more comprehensive.

As for the STEM method, it did not use any additional metadata information for learning spatial information of all single cells. The annotation information we used was just for data analysis purposes, not for training the model. Based on the mapping results given by STEM, the users could use multiple annotations such as cell cycle or drug sensitivity scores to facilitate different interests of their research. We have revised the manuscript to make readers better realize these facts (Line 450-453). Thanks.

4. More details of the work are needed:

a. Line 221: how did the authors build the spatial trajectory?

Reply 3.12:

We have added the description of constructing spatial trajectory in the method section.

b. In Figure 5, why are Basal KC, DiffKC, Cycling KC not present, but only nonTSK shown to represent them all?

Reply 3.13:

We were shown the major cell types in Fig. 5A. We have updated the figures showing all these KC subcluster locations in Fig. S24.

c. Line 319, the authors claimed that STEM helps to determine the spatial locations of rare cell types. This claim lacks clarity, as there's no evidence provided that indicates pDC cells are rare

Reply 3.14:

We have revised our statement, which now reads as “STEM streamlined the determination of the spatial location of low proportions cell types without the need for manual identification of signature genes”.

5. Line 352, The endothelial marker genes *Gja5* and *Adgrg6* are missing from Figure 6E

Reply 3.15:

We have added these two gene expression figures in the Supplementary Material.

6. Suggestions on visualization: For a clearer understanding of the expression trend along the zonation score line, it would be beneficial for the authors to add several arrows in Figure 6. These arrows could indicate that a transition from a low to a high zonation score represents a shift from the PV region to the CV region.

Reply 3.16:

We have added arrows in the revised Figure 6. Thank you for the suggestion.

REVIEWERS' COMMENTS:

Reviewer #1 (Remarks to the Author):

The authors have addressed my comments very well. New results provided in the supplementary figures are convincing and make this study more solid. I enjoy reading this new version of manuscript and look forward to seeing it in publication.

Reviewer #2 (Remarks to the Author):

The authors have addressed all my concerns. I don't have any other comments.

Reviewer #3 (Remarks to the Author):

I appreciate the authors' response, and the revised manuscript has addressed my previous questions and concerns. In their response, Hao et al. have added additional experiments to evaluate how STEM is robust against different spatial correlation structures and hyperparameters. Furthermore, the authors have conducted extra simulations to demonstrate STEM's performance in simulated datasets corresponding to various resolutions and noise levels.

In summary, the authors have demonstrated that the performance of the revised STEM method is relatively robust to the spatial correlation structure, hyperparameters, and spatial transcriptomics technological platforms. It also shows superior performance compared to state-of-the-art methods. We believe STEM could become a useful tool for mapping single-cell RNA-seq data to spatial space.

REVIEWERS' COMMENTS:

Reviewer #1 (Remarks to the Author):

The authors have addressed my comments very well. New results provided in the supplementary figures are convincing and make this study more solid. I enjoy reading this new version of manuscript and look forward to seeing it in publication.

Reviewer #2 (Remarks to the Author):

The authors have addressed all my concerns. I don't have any other comments.

Reviewer #3 (Remarks to the Author):

I appreciate the authors' response, and the revised manuscript has addressed my previous questions and concerns. In their response, Hao et al. have added additional experiments to evaluate how STEM is robust against different spatial correlation structures and hyperparameters. Furthermore, the authors have conducted extra simulations to demonstrate STEM's performance in simulated datasets corresponding to various resolutions and noise levels.

In summary, the authors have demonstrated that the performance of the revised STEM method is relatively robust to the spatial correlation structure, hyperparameters, and spatial transcriptomics technological platforms. It also shows superior performance compared to state-of-the-art methods. We believe STEM could become a useful tool for mapping single-cell RNA-seq data to spatial space.

Reply to all reviewers:

We thank all the reviewers for the reviewing our manuscript and for the detailed comments. The comments are helpful to improve the quality of the manuscript.